# The suspended small-particles layer in the oxygen-poor Black Sea: a proxy for delineating the effective N₂-yielding section

Rafael Rasse[1], Hervé Claustre[1], and Antoine Poteau[1]

[1]Sorbonne Université and CNRS, Laboratoire d'Océanographie de Villefranche (LOV) UMR7093, Institut de la Mer de Villefranche (IMEV), 06230, Villefranche-sur-Mer, France.

*Correspondence to*: rafael.rasse@obs-vlfr.fr; rjrasse@gmail.com

**Abstract.** The shallower oxygen-poor water masses of the ocean confine a majority of the microbial communities that can produce up to 90% of oceanic N₂. This effective N₂-yielding section encloses a suspended small-particle layer, inferred from particle backscattering ($b_{bp}$) measurements. It is thus hypothesized that this layer (hereafter, the $b_{bp}$-*layer*) is linked to microbial communities involved in N₂-yielding such as nitrate-reducing SAR11 as well as sulphur-oxidizing, anammox and denitrifyng bacteria — a hypothesis yet to be evaluated. Here, data collected by three BGC-Argo floats deployed in the Black Sea are used to investigate the origin of this $b_{bp}$-*layer*. To this end, we evaluate how the key drivers of N₂-yielding bacteria dynamics impact on the vertical distribution of $b_{bp}$ and the thickness of the $b_{bp}$-*layer*. In conjunction with published data on N₂ excess, our results suggest that the $b_{bp}$-*layer* is at least partially composed of the bacteria driving N₂ yielding for three main reasons: (1) strong correlations are recorded between $b_{bp}$ and nitrate; (2) the top location of the $b_{bp}$-*layer* is driven by the ventilation of oxygen-rich subsurface waters, while its thickness is modulated by the amount of nitrate available to produce N₂; (3) the maxima of both $b_{bp}$ and N₂ excess coincide at the same isopycnals where bacteria involved in N₂ yielding coexist. We thus advance that $b_{bp}$ and O₂ can be exploited as a combined proxy to delineate the N₂-yielding section of the Black Sea. This proxy can potentially contribute to refining delineation of the effective N₂-yielding section of oxygen-deficient zones via data from the growing BGC-Argo float network.

## 1 Introduction

Oxygen-poor water masses (O₂ < 3 μM) host the microbial communities that produce between 20-40% of oceanic N₂ mainly via heterotrophic denitrification and anaerobic oxidation of ammonium (Gruber and Sarmiento, 1997; Devries et al. 2013; Ward 2013). The shallower oxygen-poor water masses (~50-200 m) are the most effective N₂-producing section because this is where the microbial communities that condition the process mainly develop and generate up to 90% of the N₂ (Ward et al., 2009; Dalsgaard et al., 2012; Babin et al., 2014). These microbial communities include nitrate-reducing SAR11, and anammox, denitrifying, and sulphur-oxidizing bacteria (e.g. Canfield et al., 2010; Ulloa et al., 2012; Ward 2013; Tsementzi et al., 2016; Callbeck et al., 2018). It is thus important to unravel the biogeochemical parameters that trigger the accumulation of such bacteria in the ocean's oxygen-poor water masses. This information is crucial for understanding and quantifying how bacterial biomass and related N₂ yielding bacteria can respond to the ongoing expansion of oceanic regions with low oxygen (Keeling and Garcia, 2002; Stramma et al., 2008; Helm et al., 2011; Schmidtko et al., 2017). Ultimately, greater accuracy in this domain can contribute to improving mechanistic predictions on how such expansion affects the oceans' role in driving the Earth's climate by sequestering atmospheric carbon dioxide (e.g. Oschlies et al., 2018).

In oxygen-poor water masses, the biogeochemical factors that can affect the abundance of denitrifying and anammox bacteria are the levels of O₂, organic matter (OM), nitrate (NO₃⁻), ammonium (NH₄⁺), and hydrogen sulfide (H₂S) (Murray et al., 1995;

Ward et al., 2008; Dalsgaard et al., 2014; Bristow et al., 2016). Therefore, to elucidate what triggers the confinement of such
bacteria, we need to investigate how the above biogeochemical factors drive their vertical distribution, with high temporal and
vertical resolution. To this end, we should develop multidisciplinary approaches that allow us to permanently monitor the full
range of biogeochemical variables of interest in oxygen-poor water masses.
Optical proxies of tiny particles can be applied as an alternative approach to assess the vertical distribution of $N_2$-yielding
microbial communities in oxygen-poor water masses (Naqvi et al., 1993). For instance, nitrate-reducing SAR11, and
anammox, denitrifying, and sulphur-oxidizing bacteria are found as free-living bacteria (0.2-2 μm), and can be associated with
small suspended (> 2-30 μm), and large sinking (> 30 μm) particles (Fuchsman et al., 2011, 2012a, 2017; Ganesh et al., 2014,
2015). Therefore, particle backscattering ($b_{bp}$), a proxy for particles in the ~0.2-20 μm size range (Stramski et al., 1999, 2004;
Organelli et al., 2018), can serve to detect the presence of these free-living bacteria and those associated with small suspended
particles.
Time series of $b_{bp}$ acquired by biogeochemical Argo (BGC-Argo) floats highlight the presence of a permanent layer of
suspended small particles in shallower oxygen-poor water masses ($b_{bp}$-layer) (Whitmire et al., 2009; Wojtasiewicz et al., 2018).
It has been hypothesized that this $b_{bp}$-layer is linked to $N_2$-yielding microbial communities such as nitrate-reducing SAR11,
and denitrifying, anammox, and sulphur-oxidizing bacteria. However, this hypothesis has not yet been clearly demonstrated.
To address this, the first step is to evaluate: (1) potential correlations between the biogeochemical factors that control the
presence of the $b_{bp}$-layer and such arrays of bacteria ($O_2$, $NO_3^-$, OM, $H_2S$; Murray et al., 1995; Ward et al., 2008; Fuchsman et
al., 2011; Ulloa et al., 2012; Dalsgaard et al., 2014; Bristow et al., 2016), and (2) the possible relationship between the $b_{bp}$-
layer and $N_2$ produced by microbial communities.
This first step is thus essential for identifying the origin of the $b_{bp}$-layer and, ultimately, determining if BGC-Argo observations
of $b_{bp}$ can be implemented to delineate the oxygen-poor water masses where such bacteria are confined. The Black Sea appears
as a suitable area for probing into the origin of the $b_{bp}$-layer in low-oxygen waters in this way. It is indeed a semi-enclosed
basin with permanently low $O_2$ levels where $N_2$ production and related nitrate-reducing SAR11, and denitrifying and anammox
bacteria are mainly confined within a well-defined oxygen-poor zone (Kuypers et al., 2003; Konovalov et al., 2005; Kirkpatrick
et al., 2012). In addition, a permanent $b_{bp}$-layer is a typical characteristic of this region, which is linked to such microbial
communities and inorganic particles (Stanev et al., 2017, 2018, see details in section 2.0).
The goal of our study is therefore to investigate the origin of the $b_{bp}$-layer in the oxygen-poor waters of the Black Sea using
data collected by BGC-Argo floats. More specifically, we aim to evaluate, within the oxygen-poor zone, how: (1) two of the
main factors ($O_2$ and $NO_3^-$) that drive the dynamics of denitrifying and anammox bacteria, impact on the location and thickness
of the $b_{bp}$-layer, (2) $NO_3^-$ controls the vertical distribution of $b_{bp}$ within this layer, (3) temperature drives the formation of the
$b_{bp}$-layer and consumption rates of $NO_3^-$, and (4) particle content inferred from $b_{bp}$ and $N_2$ produced by microbial communities
can be at least qualitatively correlated. Ultimately, our findings allow us to infer that $b_{bp}$ can potentially be used to detect the
presence of the microbial communities that drive $N_2$ production in oxygen-poor water masses – *including nitrate-reducing*
*SAR11, and sulphur- oxidizing, denitrifying and anammox bacteria.*
**2.0. Background-nature of the small particles contributing to the $b_{bp}$-layer and their links with $N_2$ yielding**
The oxygen-poor water masses of the Black Sea are characterized by a permanent layer of suspended small particles constituted
of organic and inorganic particles (Murray et al., 1995; Kuypers et al., 2003; Konovalov et al., 2005; Kirkpatrick et al., 2012).
In the oxygen-poor ($O_2$ < 3 μM) section with detectable $NO_3^-$, and undetectable $H_2S$ levels, organic particles are mainly linked

to microbial communities involved in the production of $N_2$, and these include nitrate-reducing SAR11, and anammox, denitrifying, and sulphur-oxidizing bacteria (Kuypers et al., 2003; Lam et al., 2007; Yakushev et al., 2007; Fuchsman et al., 2011; Kirkpatrick et al., 2012). The first group listed, SAR11, provides $NO_2^-$ for $N_2$ yielding, and makes the largest contribution (20-60%) to $N_2$ yielding bacteria biomass (Fuchsman et al., 2011, 2017; Tsementzi et al., 2016). Meanwhile, the second and third groups of bacteria make a smaller contribution to microbial biomass (~10%; e.g. Fuchsman et al., 2011, 2017) but *dominate* $N_2$ yielding via anammox ($NO_2^- + NH_4^+ \rightarrow N_2 + 2H_2O$) and heterotrophic denitrification ($NO_3^- \rightarrow NO_2^- \rightarrow N_2O \rightarrow N_2$) (Murray et al., 2005; Kirkpatrick et al., 2012; Devries et al., 2013; Ward, 2013). The last group can potentially produce $N_2$ via autotrophic denitrification (e.g. $3H_2S + 4NO_3^- + 6OH^- \rightarrow 3SO_4^{2-} + 2N_2 + 6H_2O$; Sorokin, 2002; Konovalov et al., 2003; Yakushev et al., 2007). Finally, *Epsilonproteobacteria* are the major chemoautotrophic bacteria that form organic particles in the sulfidic zone (e.g. oxygen-poor section with detectable sulphide levels (> 0.3 μM) but undetectable $NO_3^-$; Coban-Yildiz et al., 2006; Yilmaz et al., 2006; Grote et al., 2008; Canfield and Thamdrup, 2009; Glaubitz et al., 2010; Ediger et al., 2019). However, it is also suggested that they can be involved in the production of $N_2$ and linked formation of organic particles in the oxygen-poor section with detectable levels of sulphide and $NO_3^-$ (see Figure 1, e.g. *Epsilonproteobacteria* Sulfurimonas acting as an autotrophic denitrifier; Glaubitz et al., 2010; Fuchsman et al., 2012b; Kirkpatrick et al., 2018).

The inorganic component is mainly due to sinking particles of manganese oxides (Mn, III, IV) that are formed due to the oxidation of dissolved Mn (II, III) pumped from the sulfidic zone (e.g. $2Mn^{2+}(l) + O_2 + 2H_2O \rightarrow 2MnO_2(s) + 4H^+$; Konovalov et al., 2003; Clement et al., 2009; Dellwig et al., 2010). Ultimately, sinking particles of manganese oxides are dissolved back to Mn (II, III), mainly via chemosynthetic bacteria that drive sulphur reduction (e.g. $HS^- + MnO_2(s) + 3H^+ \rightarrow S^0 + Mn^{2+}(l) + 2H_2O$; Jorgensen et al., 1991; Konovalov et al., 2003; Johnson, 2006; Yakushev et al., 2007; Fuschman et al., 2011; Stanev et al., 2018). Overall, these arrays of bacteria mediate the reactions described above by using electron acceptors according to the theoretical "electron tower" (e.g., $O_2 \rightarrow NO_3^- \rightarrow Mn(IV) \rightarrow Fe(III) \rightarrow SO_4^{2-}$; Stumm and Morgan, 1970; Murray et al., 1995; Canfield and Thamdrup, 2009). Therefore, the vertical distributions of $NO_3^-$, $N_2$ excess, and content of small particles are driven by the reactions that occur in the chemical zones of oxygen-poor water masses (e.g. nitrogenous and manganous zones, which correspond to the sections where $NO_3^-$ and Mn(IV), respectively, are predominantly used as electron acceptors; Murray et al., 1995; Konovalov et al., 2003; Yakushev et al., 2007; Canfield and Thamdrup, 2009; see also sections 4.2 and 4.3).

**3 Methods**

**3.1 Bio-optical and physicochemical data measured by BGC-Argo floats**

We used data collected by three BGC-Argo floats that profiled at a temporal resolution of 5-10 days in the first 1000 m depth of the Black Sea from December 2013 to July 2019 (Figure 1). These floats — allocated the World Meteorological Organization (WMO) numbers 6900807, 6901866, and 7900591 — collected 239, 301, and 518 vertical profiles, respectively. BGC-Argo float 6901866 was equipped with four sensors: (1) a SBE-41 CP conductivity-T-depth sensor (Sea-Bird Scientific), (2) an Aanderaa 4330 optode (serial number:1411; $O_2$ range: 0-1000 μM, with an accuracy of 1.5%), (3) a WETLabs ECO Triplet Puck, and (4) a Satlantic Submersible Ultraviolet Nitrate Analyzer (SUNA). These sensors measured upward profiles of: (1) temperature (T), conductivity, and depth, (2) dissolved oxygen ($O_2$), (3) chlorophyll fluorescence, total optical backscattering (particles + pure seawater) at 700 nm and fluorescence by Colored Dissolved Organic Matter, and (4) nitrate ($NO_3^-$; detection limit of ~0.5 μM with T/salinity correction processing) and bisulfide ($HS^-$, detection limit of ~0.5 μM; Stanev et al., 2018). Floats 6900807 and 7900591 were equipped with only the first three sensors.

Raw data of fluorescence and total backscattering were converted into Chlorophyll concentration (*chl*) and particle
backscattering (*$b_{bp}$*) following standard protocols, respectively (Schmechtig et al., 2014, 2015). Spike signals in vertical
profiles of *chl* and *$b_{bp}$* and due to particle aggregates were removed by using a median filter with a window size of three data
points (Briggs et al., 2011). $NO_3^-$, $HS^-$ and $O_2$ data were processed following BGC-Argo protocols (Bittig and Körtzinger,
2015; Johnson et al., 2018; Thierry et al., 2018). Sampling regions covered by the three floats encompassed most of the Black
Sea area (Figure 1, and Appendix A). However, we only used data collected during periods without a clear injection of small
particles derived from the productive layer and Bosporus plume (e.g. advection of water masses, Stanev et al., 2017). This
restriction allowed us to focus on the *in-situ* 1D processes driving local formation of the *$b_{bp}$-layer*, with minimal interference
from any possible external sources of small particles.
We only describe the time series of data collected by float 6901866 because this was the only float carrying a $NO_3^-$/$HS^-$ sensor.
Data acquired by floats 6900807 and 7900591 are described in Appendix A, and nevertheless used as complementary data to
those of float 6901866 to corroborate: (1) qualitative correlations between $O_2$ levels and the location of the *$b_{bp}$-layer*, and (2)
consistency in the location of the *$b_{bp}$* maximum within the *$b_{bp}$-layer*.
**3.2 Defining the oxygen-poor zone, mixed layer depth, and productive layer**
We used $O_2$ and $NO_3^-$ to respectively define the top and bottom isopycnals of the oxygen-poor zone where denitrifying and
anammox bacteria are expected to be found. To set the top isopycnal, we applied an $O_2$ threshold of ~3 µM because denitrifying
and anammox bacteria seem to tolerate $O_2$ concentrations beneath this threshold (Jensen et al., 2008; Dalsgaard et al., 2014;
Babbin et al., 2014). The bottom isopycnal was defined as the deepest isopycnal at which $NO_3^-$ was detected by the SUNA
sensor (0.23 ± 0.32 µM). $NO_3^-$ was used to set this isopycnal because heterotrophic denitrification and subsequent reactions
cannot occur without $NO_3^-$ (Lam et al., 2009; Bristow et al., 2017). $HS^-$ was not used to delimit the bottom of this zone because
the maximum concentration of $HS^-$ that denitrifying and anammox bacteria tolerate is not well established (Murray et al., 1995;
Kirkpatrick et al., 2012; see also section 4.1).
Mixed layer depth (MLD) was computed as the depth at which density differed from 0.03 kg m$^{-3}$ with respect to the density
recorded at 1m depth (de Boyer Montégut et al., 2004). We used *chl* to define the productive layer where living phytoplankton
were present and producing particulate organic carbon. The base of this layer was set as the depth at which *chl* decreased
below 0.25 mg m$^{-3}$. This depth was used only as a reference to highlight the periods when surface-derived small particles were
clearly injected into the oxygen-poor zone.
**3.3 Complementary cruise data on $N_2$ excess and $NO_3^-$**
Published data on $N_2$:Ar ratios and $NO_3^-$ collected at the southwest of the Black Sea in March 2005 (Fuchsman et al., 2008,
2019) were exploited to complement discussion of our results. $N_2$ produced by anaerobic microbial communities ($N_2$ excess,
µM) was estimated from $N_2$:Ar ratios and argon concentrations at atmospheric saturation (Hamme and Emerson, 2004). $N_2$
excess data were used to: (1) describe the oxygen-poor zone where $N_2$ is expected to be predominantly produced, and (2)
highlight qualitative correlations between $N_2$ excess, the location of the *$b_{bp}$-layer*, and vertical distribution of small particles
within the *$b_{bp}$-layer*.

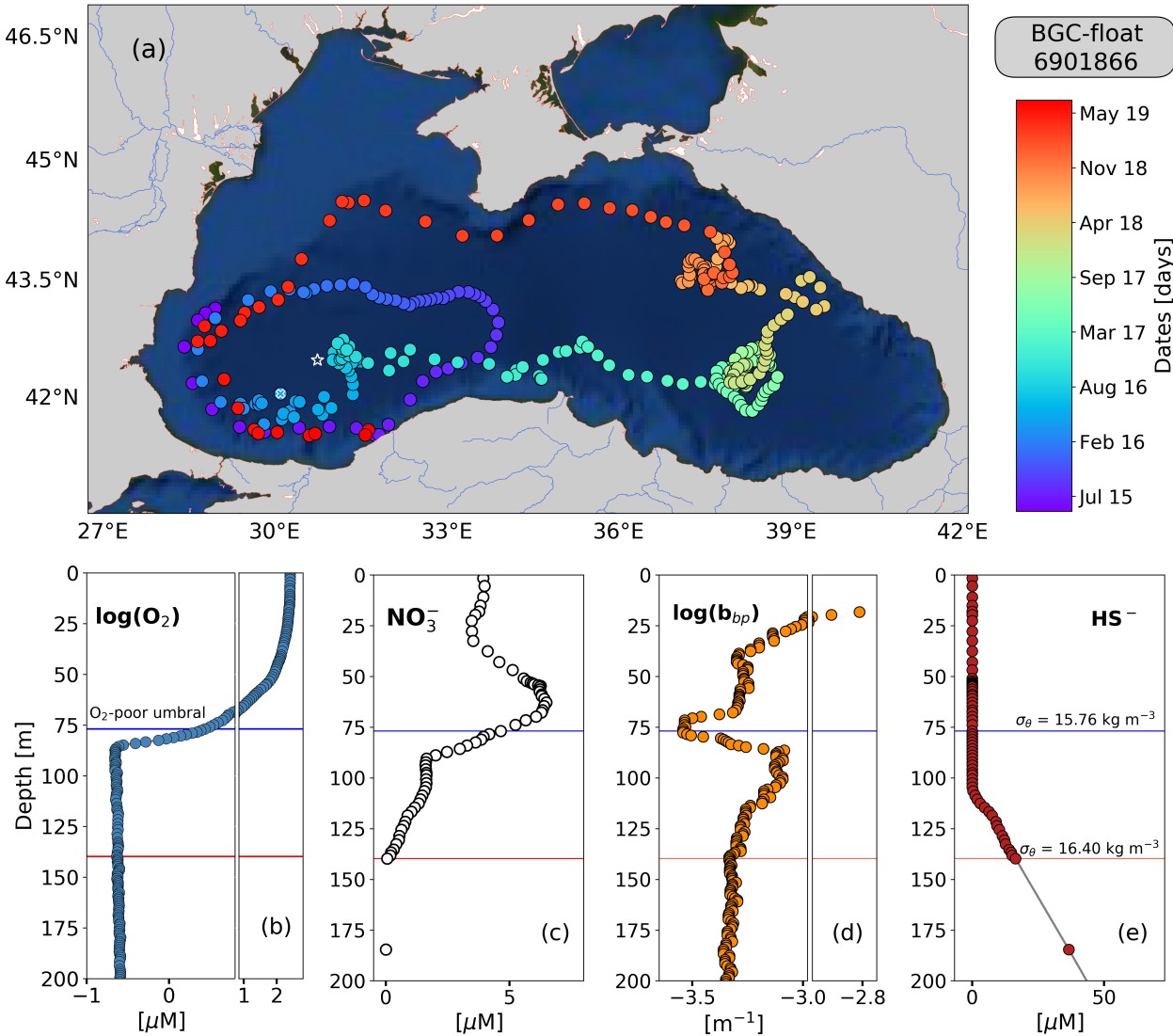

Figure 1: (a) Sampling locations of float 6901866 between May 2015 and July 2019. Colored circles indicate the date (color bar) for a given profile. The white star in (a) marks the sampling site of the cruise (March 2005). The white *x* in (a) highlights the float location on 6th April 2016. Float profiles of (b) log($O_2$), (c) $NO_3^-$, (d) log($b_{bp}$), and (e) $HS^-$ collected on 24th November 2018.

**4 Results and discussion**

**4.1 Description of the oxygen-poor zone**

The top and bottom of the oxygen-poor zone are located around the isopycnals (mean ± standard deviation) 15.79 ± 0.23 kg m$^{-3}$ and 16.30 ± 0.09 kg m$^{-3}$, respectively. The two isopycnals therefore delimit the oxygen-poor water masses where nitrate-reducing SAR11, and denitrifying, anammox, and sulphur-oxidizing bacteria are expected to be found (zone hereafter called the $OP_{D-A}$, Figure 2; Kuypers et al., 2003; Lam et al., 2007; Yakushev et al., 2007; Fuschman et al., 2011; Kirkpatrick et al., 2012). The top location of the $OP_{D-A}$ shows large spatial-temporal variability ranging between 80-180 m (or $\sigma_\theta$ between 15.5-15.9 kg m$^{-3}$, Figure 2). Similarly, the $OP_{D-A}$ thickness varies between 30-80 m, which corresponds to a $\sigma_\theta$ separation of ~0.50 kg m$^{-3}$. The bottom of the $OP_{D-A}$ is slightly sulfidic ($HS^-$ = 11.4 ± 3.53 µM, n = 86) and deeper than suggested (e.g. $\sigma_\theta$ = 16.20

160  kg m$^{-3}$, and H$_2$S ≤ 10 nM, Murray et al., 1995). However, our results coincide with the slightly sulfidic conditions of the deepest

161  isopycnal at which anammox bacteria can be still recorded (σ$_θ$ = 16.30 kg m$^{-3}$, and H$_2$S ≥10 μM; Kirkpatrick et al., 2012).

Figure 2: Time series of: (a) Salinity (S), (b) O$_2$, (c) NO$_3^-$, (d) log($b_{bp}$), and (e) HS$^-$. The blue lines in (a) and (b) indicate the mixed layer depth. The red lines in (a), (b) and (d) show the base of the productive region. The isopycnals 15.79 kg m$^{-3}$ and 16.30 kg m$^{-3}$ describe the top and bottom of the oxygen-poor zone (OP$_{D-A}$), respectively. SU, A, W, and SP stand for summer, autumn, winter, and spring, respectively. The colored horizontal line in (b) indicates the sampling site for a given date (Figure 1). The horizontal white lines in (d) are the profiles used to: (1) delimit the OP$_{D-A}$, and (2) compute correlations between $b_{bp}$, NO$_3^-$, and T within the OP$_{D-A}$.

**4.2 NO$_3^-$, O$_2$, and MnO$_2$ as key drivers of the thickness and location of the suspended small-particle layer**

The permanent $b_{bp}$-*layer* is always confined within the two isopycnals that delimit the OP$_{D-A}$ (Figure 2). It follows that the
thickness and top location of this layer demonstrate the same spatial and temporal variability as the one described for the OP$_{D-}$
$_A$ (Figure 2 and Appendix A). This correlation indicates that variations in the thickness and top location of the $b_{bp}$-layer are
partially driven, respectively, by: (1) the amount of $NO_3^-$ available to produce $N_2$ inside the $OP_{D-A}$ via the set of bacteria
communities involved, and (2) downward ventilation of oxygen-rich subsurface waters (Figure 2 and Appendix A).
$NO_3^-$ and $O_2$ are two of the key factors that modulate the presence of: (1) denitrifying and anammox bacteria working in
conjunction with nitrate-reducing SAR11 (Fuschman et al., 2011; Ulloa et al., 2012; Tsementezi et al., 2016; Bristow et al.,
2017), and probably with chemoautotrophic ammonia-oxidizing bacteria (in this case, only with anammox, e.g. γAOB; Ward
and Kilpatrick, 1991; Lam et al., 2007), and (2) sulphur-oxidizing bacteria (e.g. SUP05 and potentially *Epsilonproteobacteria*
Sulfurimonas; Canfield et al., 2010; Glaubitz et al., 2010; Fuschman et al., 2011, 2012b; Ulloa et al., 2012; Kirkpatrick et al.,
2018). Therefore, the results described above highlight that at least a fraction of the $b_{bp}$-layer should be due to this array of
bacteria. This notion is supported by three main observations. Firstly, the top location of the $b_{bp}$-layer is driven by the intrusion
of subsurface water masses (S ≤ 20.36 ± 0.18 psu) with $O_2$ concentrations above the levels tolerated by denitrifying and
anammox bacteria ($O_2 \geq 3\,\mu M$, Jensen et al., 2008; Babbin et al., 2014; Figure 2). As a result, in regions where $O_2$ is ventilated
to deeper water masses, the top location of the $b_{bp}$-layer is also deeper. The contrary is observed when $O_2$ ventilation is
shallower (Figure 2 and Appendix A). Secondly, nitrate-reducing SAR11, and denitrifying, anammox, and sulphur-oxidizing
bacteria reside between the isopycnals 15.60-16.30 kg m$^{-3}$ (Fuchsman et al., 2011; 2012a; Kirkpatrick et al., 2012), while the
$b_{bp}$-layer is formed between isopycnals ~15.79-16.30 kg m$^{-3}$. We can thus infer coexistence of such bacteria between the
coincident isopycnals where the $b_{bp}$-layer is generated. Thirdly, $NO_3^-$ declines from around isopycnal 15.79 kg m$^{-3}$ to the
isopycnal 16.30 kg m$^{-3}$ due to the expected $N_2$ production via the microbial communities involved (Figures 2-3, and Kirkpatrick
et al., 2012).
The ventilation of subsurface $O_2$ is also key in driving the depth at which $MnO_2$ is formed ($O_2 \leq 3\text{-}5\,\mu M$; Clement et al., 2009),
and can thus contribute to setting the characteristics of the $b_{bp}$-layer via its subsequent accumulation and dissolution
(Konovalov et al., 2003; Clement et al., 2009; Dellwig et al., 2010). Thus, in regions where subsurface $O_2$ (e.g. $O_2 \geq 3\text{-}5\,\mu M$,
and S ≤ 20.36 ± 0.18 psu) is ventilated to deeper water masses, both the formation of $MnO_2$ and top location of the $b_{bp}$-layer
can be expected to be deeper, and vice versa (Figure 2). Finally, the dissolution of $MnO_2$ should also influence the thickness
of the $b_{bp}$-layer because it occurs just beneath the maxima of the optical particles inside this *layer* (Konovalov et al., 2006; see
the explanation in section 4.3).
Overall, the qualitative evidence presented above points out that particles of $MnO_2$ as well as nitrate-reducing SAR11, and
denitrifying, anammox, and sulphur-oxidizing bacteria, appear to define the characteristics of the $b_{bp}$-layer (Johnson, 2006;
Konovalov et al., 2003; Fuchsman et al., 2011, 2012b; Stanev et al., 2018). This observation leads us to argue, in the next
section, that the $b_{bp}$-layer is partially composed of the main group of microbial communities involved in $N_2$ yielding, as well
as of $MnO_2$.
**4.3 Role of the removal rate of $NO_3^-$, $MnO_2$, and temperature in the vertical distribution of small particles**
We propose that the removal rate of $NO_3^-$ is a key driver of the vertical distribution of small particles and $N_2$ excess within the
$OP_{D-A}$. This is because the vertical profiles of small particles and of $N_2$ excess are qualitatively similar, and both profiles are
clearly related to the rate at which $NO_3^-$ is removed from the $OP_{D-A}$ (Figures 3-4). For instance, maxima of $N_2$ excess and $b_{bp}$
coincide around the isopycnal 16.11 ± 0.11 kg m$^{-3}$ (Figure 3; Konovalov et al., 2005; Fuchsman et al., 2008, 2019). At this
isopycnal, the mean concentration of $NO_3^-$ is 1.19 ± 0.53 μM. We thus propose that this $NO_3^-$ threshold value splits the $OP_{D-A}$
in two sub-zones with distinctive biogeochemical conditions (e.g. nitrogenous and manganous zones; Canfield and Thamdrup,
2009). Ultimately, these two different sets of conditions drive the rates at which $NO_3^-$ and small particles are removed and
formed within the $OP_{D-A}$, respectively (Figure 3, and explanation below).
The first sub-zone is thus located between the top of the $OP_{D-A}$ ($\sigma_\theta$ = 15. 79 kg m$^{-3}$) and around the isopycnal 16.11 kg m$^{-3}$.
Here, removal rates of $NO_3^-$ (-0.16 ± 0.10 μM m$^{-1}$, Figure 4) are likely to be boosted by: (1) high content of organic matter
(dissolved organic carbon = 122 ± 9 μM, Margolin et al., 2016) and $NO_3^-$ (≥ 1.19 ± 0.53 μM), and (2) $O_2$ levels staying between
a range that maintain the yielding of $N_2$ (0.24 ± 0.04 μM ≥ $O_2$ ≤ 2.8± 0.14 μM, n = 100, the means of the minima and maxima
of $O_2$, respectively, in the first sub-zone) and promote the formation of $MnO_2$ (e.g. maximum of Mn(II) oxidation is at $O_2$ levels
~0.2 μM; Clement et al., 2009). Consequently, the formation of biogenic and inorganic small particles (and related $N_2$ excess)
increases from the top of the $OP_{D-A}$ to around the isopycnal 16.11 kg m$^{-3}$ (Figure 3). This hypothesis is: (1) in part confirmed
by significant and negative power-law correlations between the suspended small-particle content and $NO_3^-$ in this sub-zone
(Figure 3), and (2) in agreement with the progressive accumulation of $MnO_2$ from around isopycnal 15.8 kg m$^{-3}$ to the isopycnal
16.10 kg m$^{-3}$ (e.g. Konovalov et al., 2006).
The second sub-zone is located between isopycnal 16.11 kg m$^{-3}$ and the bottom of the $OP_{D-A}$ ($\sigma_\theta$ = 16.30 kg m$^{-3}$, Figure 3).
Here, $NO_3^-$ is low (≤ 1.19 ± 0.53 μM) and $O_2$ is relatively constant (0.23 ± 0.02 μM, n= 2284, mean of $O_2$ calculated in the
second sub-zone for all profiles), or lower than the minimum of $O_2$ recorded by this sensor (0.22 ± 0.02 μM, n = 89). These
constant (or lower) levels of $O_2$ roughly correspond to those at which anammox and heterotrophic denitrification are inhibited
by ~50% (0.21 μM, and 0.81 μM, respectively; Dalsgaard et al., 2014). In addition, low levels of $NO_3^-$ necessarily promotes
the microbial use of Mn(IV) as an electron acceptor, ultimately dissolving the particles of $MnO_2$ into Mn(II) (e.g. manganous
zone; Konovalov et al., 2006; Yakushev et al., 2007; Canfield and Thamdrup, 2009). As a result, this sub-zone exhibits a
decline in removal rates of $NO_3^-$ (-0.04 ± 0.01 μM m$^{-1}$, Figure 4) along with inhibited formation of biogenic small particles and
dissolution of $MnO_2$. Ultimately, both the content of small particles and related $N_2$ excess decrease from around isopycnal
16.11 kg m$^{-3}$ to the bottom of the $OP_{D-A}$ (Figure 3). These results are in agreement with: (1) significant and positive exponential
correlations computed between the small-particle content inferred from $b_{bp}$ and $NO_3^-$ within this sub-zone (Figure 3), and (2)
the overlap of nitrogenous and manganous zones in this sub-zone because the content of $MnO_2$ particles and dissolved Mn(II)
concurrently declines and increases just beneath the isopycnal 16.11 kg m$^{-3}$, respectively (e.g. Murray et al., 1995; Konovalov
et al., 2003, 2005, 2006; Yakushev et al., 2007; Canfield and Thamdrup, 2009).
Strong-positive linear correlations are also recorded between $b_{bp}$ and T in the first sub-zone of the $OP_{D-A}$ (Figure 4). This  is
likely to indicate that the formation of small particles is sensitive to very tiny increments in T (0.003 ± 0.001 °C m$^{-1}$, n = 133).
We thus infer a tendency for the decline rates of $NO_3^-$ and related production of $N_2$ to increase with T. This hypothesis is at
least partially supported by the significant correlation between $NO_3^-$ decline rates and T increase rates in this sub-zone (Figure
4). Within the second sub-zone, T continues increasing while $b_{bp}$ decreases, likely due to inhibition of the formation of small
particles for the reasons described above (Figure 4). These observations suggest that the production of small particles is likely
to have first- and second-order covariations, with $NO_3^-$ and T, respectively — a likelihood backed up by a lack of correlation
between $NO_3^-$ decline rates and T increase rates in this sub-zone (Figure 4). Finally, more information is needed to investigate
the physical and/or biogeochemical processes driving the correlation between the increase rates of T, and declines rates of
$NO_3^-$ in the first sub-zone. This is however out of the scope of our study.

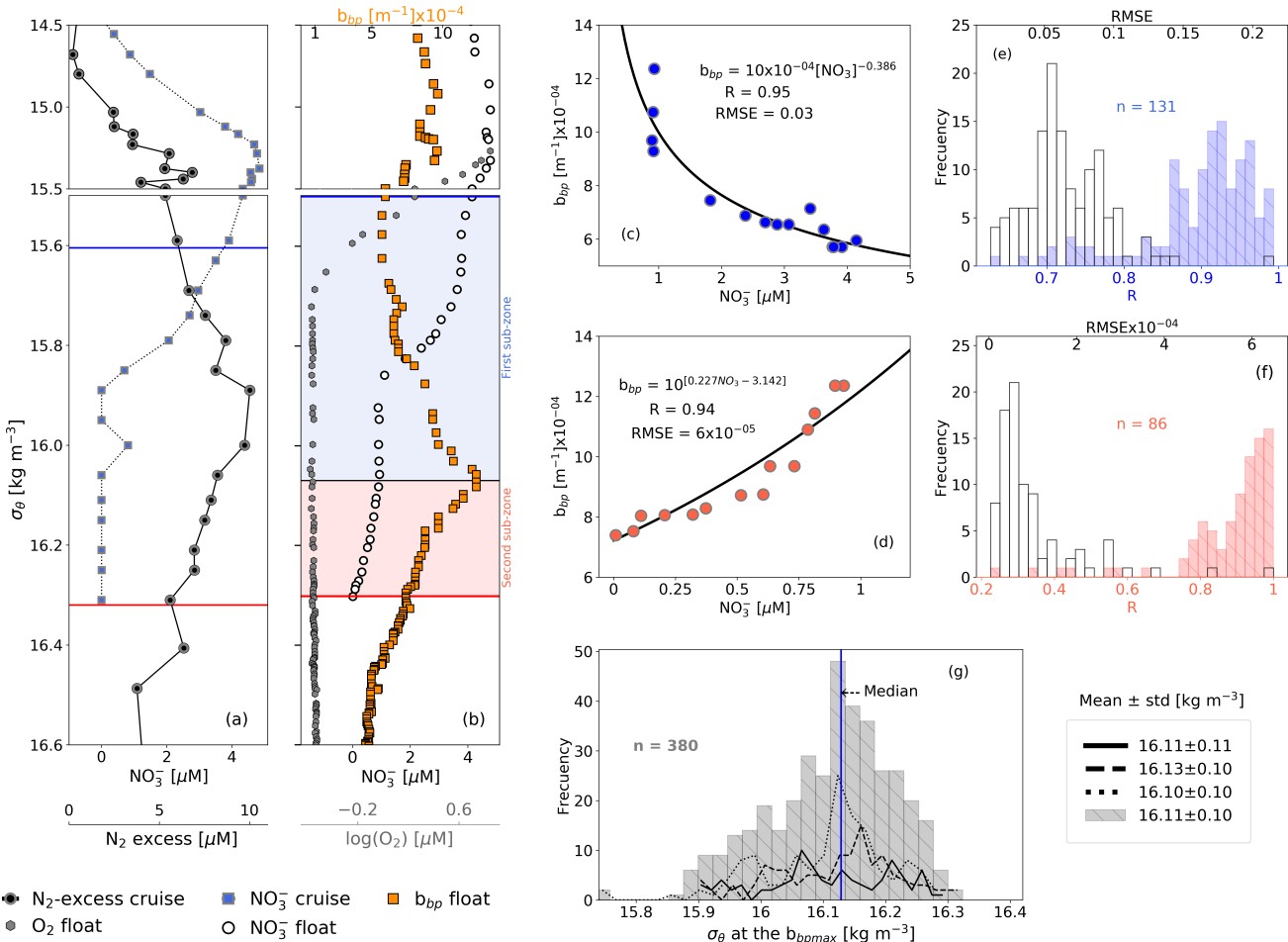

**Figure 3:** (a) Cruise profiles of $NO_3^-$, and $N_2$ excess, collected in March 2005 (Fuchsman et al., 2019). (b) Float profiles of $NO_3^-$, $b_{bp}$, and $\log(O_2)$ measured on 6th April 2016. Profiles in (a) and (b) were conducted at the northwest of the basin (see Figure 1). The top and bottom of the $OP_{D-A}$ are described in (a) and (b) as horizontal blue and red lines, respectively. The $b_{bp}$ maximum is the horizontal black line in (b). The first and second sub-zone of the $OP_{D-A}$ are respectively highlighted in (b) as blue and red squares. $NO_3^-$ vs $b_{bp}$ in (c) the first, and (d) the second sub-zone, of the float profile in (b). The number of data points visualized in (c) is lower than in (b) for the first sub-zone because $b_{bp}$ and $NO_3^-$ are not always recorded at the same depths. (e) Frequency distributions of correlation coefficients (R, blue bars), and root mean square errors (RMSE, white bars) for $NO_3^-$ vs $b_{bp}$ in the first sub-zone. (f) Same as (e) but for the second sub-zone. (g) Frequency distributions of the isopycnals at which $b_{bp}$ maxima are found within the $OP_{D-A}$. Dotted, dashed, and solid black lines in (g) are data collected by floats 7900591, 6901866, and 6900807, respectively. Gray bars include all data.

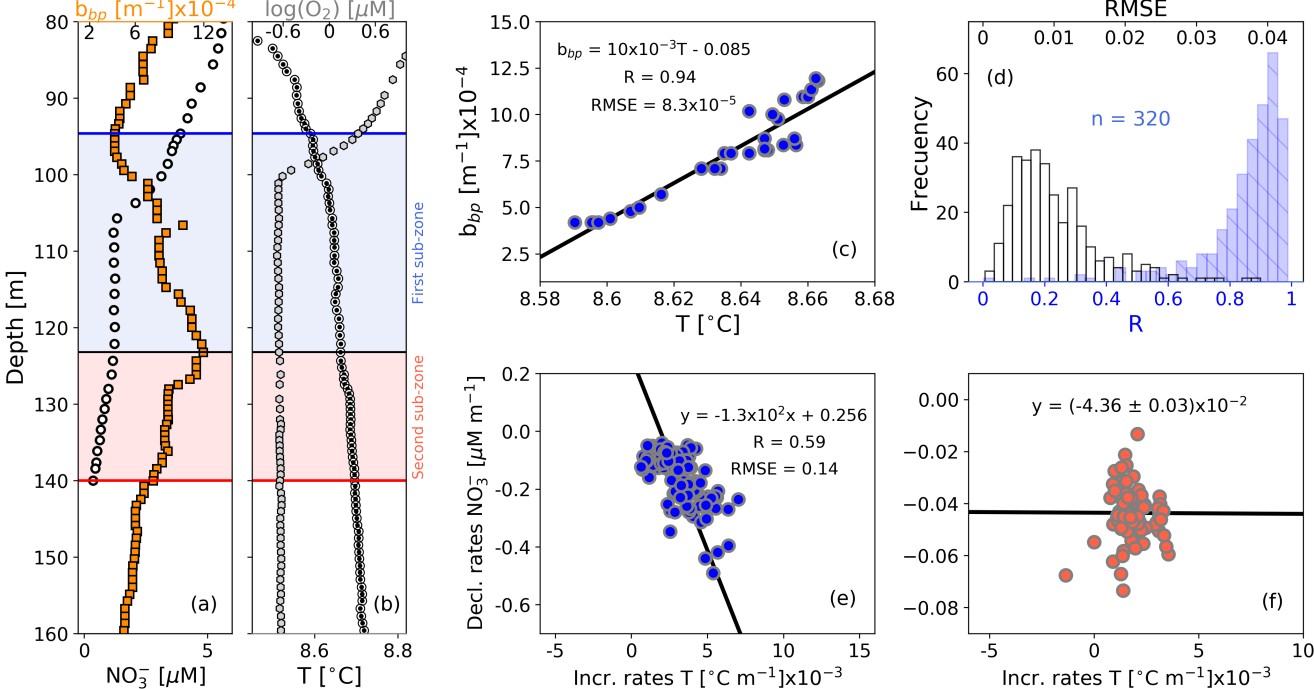

**Figure 4: Float profiles of (a) NO₃⁻, and $b_{bp}$, and (b) T and log(O₂) collected on 10th September 2017. Horizontal blue and red lines in (a) and (b) are the top and bottom of the $OP_{D-A}$. The $b_{bp}$ maximum is indicated in (a) and (b) as horizontal black lines. The first and second sub-zones of the $OP_{D-A}$ are respectively highlighted in (a) and (b) as blue and red squares. (c) $b_{bp}$ vs T for the first sub-zone of the profile in (b). (d) Frequency distributions of correlation coefficients (R, blue bars), and root mean square errors (RMSE, white bars), for $b_{bp}$ vs T in the first sub-zone, including data collected by the three floats. Decrease rates of NO₃⁻ vs increase rates of T in (e) the first and (f) the second sub-zone.**

To summarize, BGC-Argo float data combined with a proxy of $N_2$ production suggest that in regions without the Bosporus plume influence, the $b_{bp}$-*layer* systematically tracks and delineates the *effective* $N_2$-yielding section independently of: (1) the biogeochemical mechanisms driving $N_2$ yielding, and (2) the contribution that $MnO_2$ and other microorganisms can be expected to make to the formation of the $b_{bp}$-*layer* (e.g. Lam et al., 2007; Fuchsman et al., 2011; 2012a; Kirkpatrick et al., 2018). It is thus finally inferred that this $b_{bp}$-*layer* is *at least partially* composed of the predominant anaerobic microbial communities involved in the production of $N_2$, such as *nitrate-reducing SAR11, and anammox, denitrifying, and sulphur-oxidizing* bacteria. These results also suggest that $N_2$ production rates can be highly variable in the Black Sea because the characteristics of the $b_{bp}$-*layer* show large spatial-temporal variations driven by changes in NO₃⁻ and O₂ (Figures 2 and 4). Finally, we propose that $b_{bp}$ and O₂ can be exploited as a combined proxy for defining the $N_2$-producing section of the oxygen-poor Black Sea. We consider that this combined proxy can delineate the top and base of this section, by applying an O₂ threshold of 3.0 μM, and the bottom isopycnal of the $b_{bp}$-*layer*, respectively. This section should thus be linked to free-living bacteria (0.2-2 μm), and those associated with small suspended particles (> 2-20 μm), as well as to small inorganic particles (0.2-20 μm).

### 4.4 New perspectives for studying $N_2$ losses in ODZs

The conclusions and inferences of this study, especially those related to the origin and drivers of the $b_{bp}$-*layer*, primarily apply to the Black Sea. However, these findings may also have a wider application. In particular, the shallower water masses of oxygen-deficient zones (ODZs) are similarly characterized by the formation of a layer of suspended small particles that can

be optically detected by $b_{bp}$ and the attenuation coefficients of particles (Spinrad et al., 1989; Naqvi et al., 1993; Whitmire et al., 2009). This layer is mainly linked to $N_2$-yielding microbial communities because: (1) its location coincides with the maxima of $N_2$ excess, microbial metabolic activity, and nitrite ($NO_2^-$, the intermediate product of denitrification-anammox that is mainly accumulated in the $N_2$-yielding section, Spinrad et al., 1989; Naqvi et al., 1991, 1993; Devon et al., 2006; Chang et al., 2010, 2012; Ulloa et al., 2012; Wojtasiewicz et al., 2018), and (2) $MnO_2$ is not accumulated as in the Black Sea (Martin and Knauer, 1984; Johnson et al., 1996; Lewis and Luther, 2000). Therefore, our findings suggest that highly resolved vertical profiles of $b_{bp}$ and $O_2$ can potentially be used as a combined proxy to define the *effective* $N_2$-production section of ODZs. Such definition can be key to better-constrained global estimates of $N_2$ loss rates because it can allow us to: (1) accurately predict the oxygen-poor water volume where around 90% of $N_2$ is produced in the ODZ core (Babin et al., 2014), and (2) evaluate how the location and thickness of the $N_2$-yielding section vary due to changes in the biogeochemical factors that modulate anammox and heterotrophy denitrification.

Global estimates of $N_2$ losses differ by 2-3 fold between studies (e.g. 50-150 Tg N $yr^{-1}$, Codispoti et al., 2001; Bianchi et al., 2012, 2018; DeVries et al., 2012; Wang et al., 2019). These discrepancies are caused in part by inaccurate estimations of the oxygen-poor volume of the $N_2$-production section. Other sources of uncertainties arise from the methods applied to estimate the amount of POC that fuels $N_2$ production. For instance, POC fluxes and their subsequent attenuation rates are not well resolved because they are computed respectively from satellite-based primary-production algorithms and generic power-law functions (Bianchi et al., 2012, 2018; DeVries et al., 2012). POC-flux estimates based on these algorithms visibly exclude: (1) POC supplied by zooplankton migration (Kiko et al., 2017; Tutasi and Escribano, 2020), (2) substantial events of POC export decoupled from primary production (Karl et al., 2012), and (3) the role of small particles derived from the physical and biological  fragmentation of larger ones (Karl et al., 1988; Briggs et al., 2020), which are more efficiently remineralized by bacteria in ODZs (Cavan et al., 2017). In addition, these estimates do not take into consideration the inhibition effect that $O_2$ intrusions may have on $N_2$-yield rates (Whitmire et al., 2009; Ulloa et al., 2012; Dalsgaard et al., 2014; Peters et al., 2016; Margolskee et al., 2019).

Overall, mechanistic predictions of $N_2$ losses misrepresent the strong dynamics of the biogeochemical and physical processes that regulate them. Consequently, it is still debated whether the oceanic nitrogen cycle is in balance or not (Codispoti, 2007; Gruber and Galloway, 2008; DeVries et al., 2012; Jayakumar et al., 2017; Bianchi et al., 2018; Wang et al., 2019). The subsiding uncertainty points to a compelling need for alternative methods that allow accurate refinement of oceanic estimations of $N_2$ losses.

Our study supports the proposition that robotic observations of $b_{bp}$ and $O_2$ can be used to better delineate the $N_2$-yielding section at the appropriate spatial (e.g. vertical and regional) and temporal (e.g. event, seasonal, interannual) resolutions. In addition, POC fluxes and $N_2$ can be simultaneously quantified using the same float technology (BGC-Argo, Bishop et al., 2009; Dall'Olmo and Mork, 2014; Reed et al., 2018; Boyd et al., 2019; Estapa et al., 2019; Rasse and Dall'Olmo, 2019). These robotic measurements can contribute to refining global estimates of $N_2$ losses by better constraining both the oxygen-poor section where $N_2$ is produced, and POC fluxes that fuel its loss. Ultimately, $O_2$ intrusions into the $N_2$-yielding section can potentially be quantified by BGC-Argo floats to assess their regulatory effect on $N_2$ losses.

**Conclusions**

Our results along with those from previous studies suggest that the $b_{bp}$-*layer* of the oxygen-poor Black Sea is at least partially composed of nitrate-reducing SAR11, and anammox, denitrifying, and sulphur-oxidizing bacteria. The location and thickness of this layer show strong spatial-temporal variability, mainly driven by the ventilation of oxygen-rich subsurface waters, and

nitrate available to generate N$_2$, respectively. Such variations in the characteristics of the $b_{bp}$-*layer* highlight that N$_2$-production
rates can be highly variable in the Black Sea. We therefore propose that high resolution measurements of O$_2$ and $b_{bp}$ can
potentially be exploited as a combined proxy to delineate the *effective* N$_2$-yielding section of ODZs. This proposition is in part
supported by evidence that the $b_{bp}$-*layer* and a majority of N$_2$-yielding microbial communities are both confined in the
shallower oxygen-poor water masses of ODZs. We however recommend investigation into the key biogeochemical drivers of
the $b_{bp}$-*layer* for each ODZ. This information will be critical for validating the applicability of the $b_{bp}$-*layer* in assessing spatial-
temporal changes in N$_2$ production.
Finally, it is evident that BGC-Argo float observations can acquire essential proxies of N$_2$ production and associated drivers
at appropriate spatial and temporal resolutions. The development of observation-modeling synergies therefore holds the
potential to deliver an unprecedented view of N$_2$-yielding drivers if robotic observations become an integrated part of model
validation. Ultimately, this approach could prove essential for reducing present uncertainties in the oceanic N$_2$ budget.

## Appendix A: Supplementary Figures

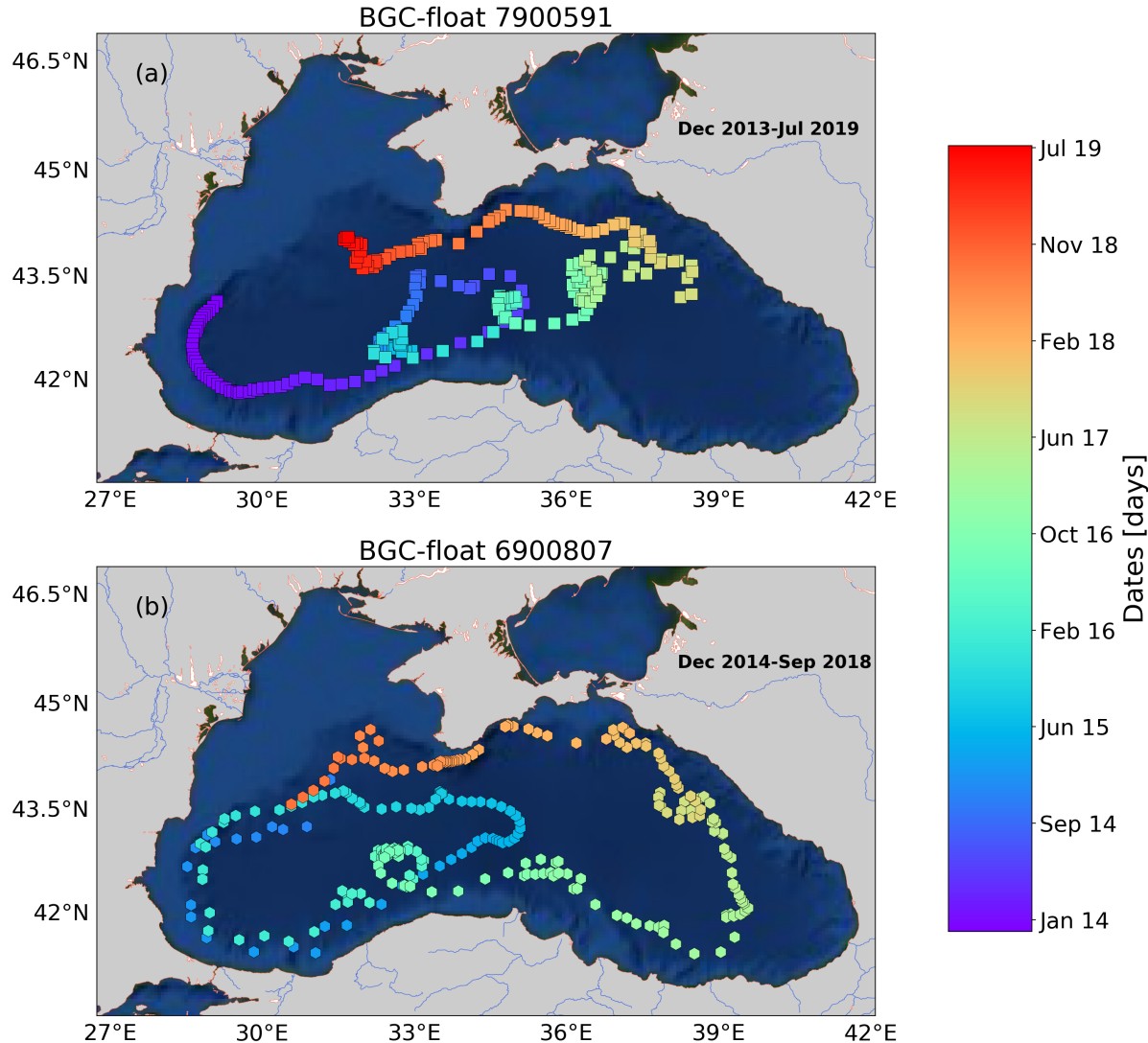

**Figure A1: Sampling locations of floats (a) 7900591 and (b) 6900807 between December 2013 and July 2019. Colored squares and hexagons indicate the date (colorbar) for a given profile of floats 6900807 and 7900591, respectively.**

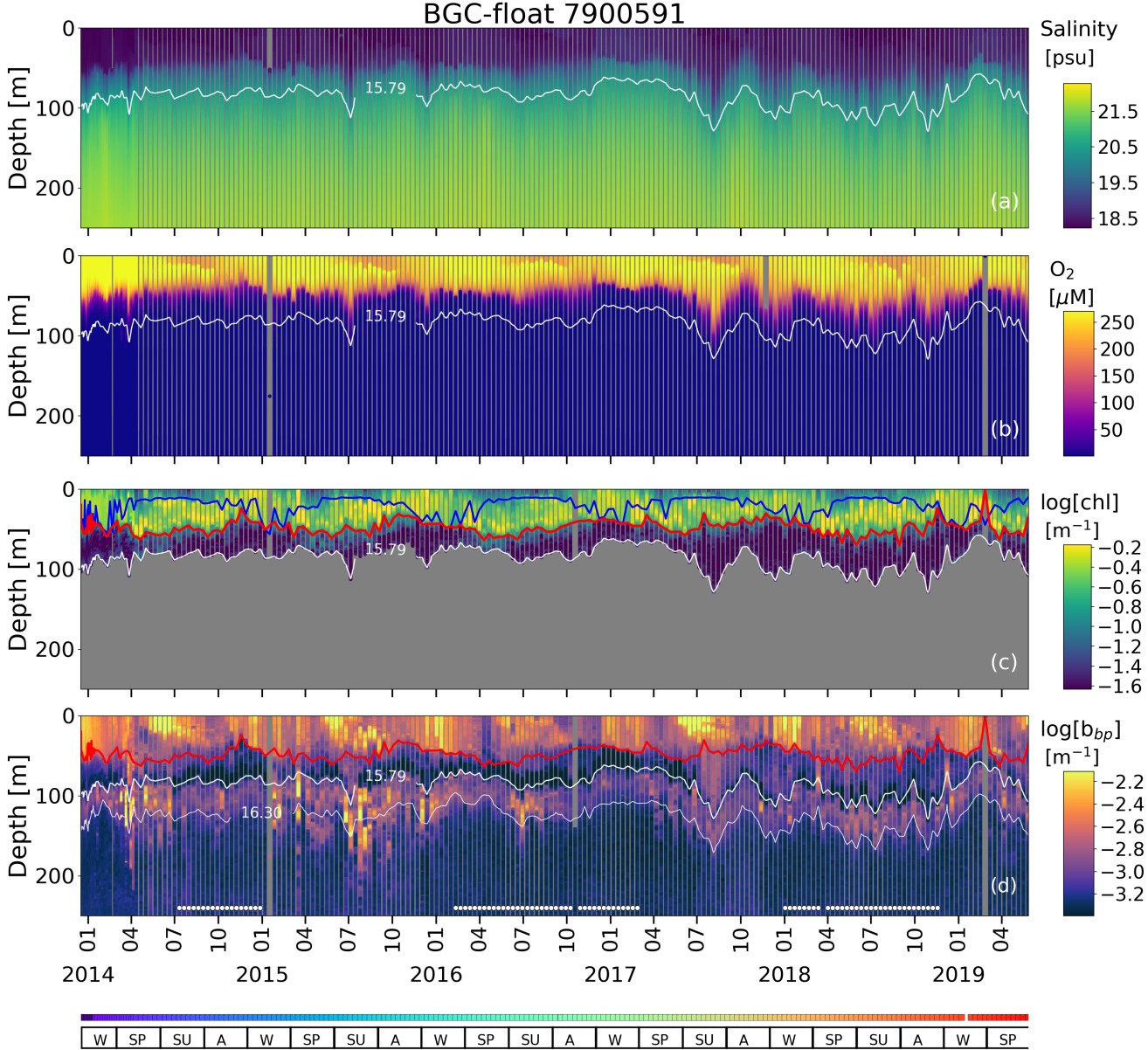


**Figure A2: Time series of (a) S, (b) O₂, (c) log(*chl*), and (d) log(*b*$_{bp}$) for float 7900591. The blue line in (c) indicates the**
**mixed layer depth. The red lines in (c) and (d) show the base of the productive region. The isopycnals 15.79 kg m$^{-3}$ and**
**16.30 kg m$^{-3}$ describe the top and bottom of the oxygen-poor zone (*OP*$_{D-A}$), respectively. SU, A, W, and SP stand for**
**summer, autumn, winter, and spring, respectively. The colored horizontal line at the bottom indicates the sampling site**
**for a given date (Figure S1). The horizontal white lines in (d) are the profiles used to: (1) delimit the *SO*$_{D-A}$, and (2) find**
**the isopycnals at which *b*$_{bp}$ is maximum in the *SO*$_{D-A}$. *chl* is set to zero in the *SO*$_{D-A}$ due to fluorescence contamination**
**(Stanev et al., 2017).**

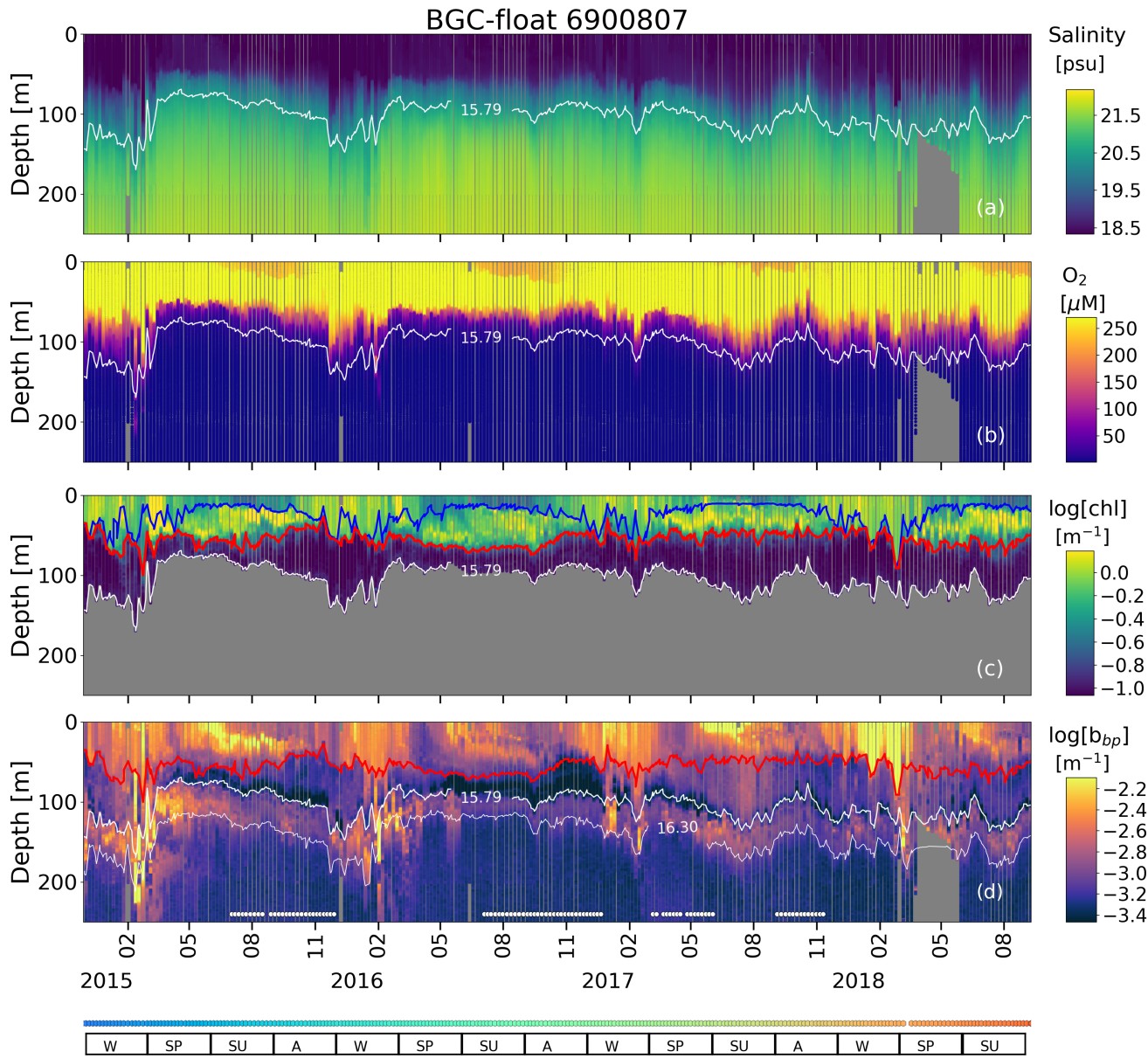

**Figure A3: Same as Figure A2 but for float 6900807**

*Data availability*. Data from Biogeochemical-Argo floats used in this study are freely available at ftp.ifremer.fr/ifremer/argo. These data were collected and made freely available by the International Argo Program and the national programs that contribute to it (http://www.argo.ucsd.edu; the Argo Program is part of the Global Ocean Observing System). Data on $N_2$:Ar ratios are freely available at https://agupubs.onlinelibrary.wiley.com/doi/abs/10.1029/2018GB006032.

*Author contributions*. R.R. conceptualized the study, wrote the original draft, and generated all figures. H.C. contributed to tuning the study's conceptualization and figures design. A.P. processed all BGC-Argo float data. R.R. and H.C. reviewed and edited the final manuscript. We finally thank Dr. Clara A. Fuchsman and the anonymous reviewer for their accurate and constructive feedback, which allowed us to significantly improve the original version of the manuscript.

*Acknowledgments*. This study was conducted in the framework of the *Noceanic* project. This project is funded by the European Union's Horizon 2020 research and innovation program under the Marie Skłodowska-Curie Individual Fellowship awarded to

Rafael Rasse (grant agreement 839062). This study is a contribution to the remOcean project (European Research Council, grant agreement 246777, Hervé Claustre).

*Competing interests*. The authors declare that they have no conflicts of interest.

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
