# Peer review of "The suspended small-particles layer in the oxygen-poor Black Sea: a proxy for delineating the effective N2-yielding section"

_Biogeosciences, 2020_

## Referee Comment (RC1) · Clara A Fuchsman (Referee) · 7 Jul 2020

In "The suspended small-particles layer in the suboxic Black Sea: a proxy for delineating the effective N2-yielding section" by Rasse et al, the authors analyze particle back scattering, oxygen, HS- and nitrate float data from the Black Sea. The authors can thus delineate the suboxic zone of the Black Sea with the float, and see productivity and export events. The authors assume that the particle back scattering data in the suboxic zone indicates the presence of anammox and heterotrophic denitrifying bacteria. This assumption may be problematic in the Black Sea where it is known that there are high manganese oxide concentrations in the suboxic zone, and it is known

that there is an organic matter maximum at the top of the sulfidic zone composed of S oxidizers. However, the data in this paper is useful. The writing just needs to be shifted.

Large Issues:

The introduction needs a section at the beginning describing the Black Sea. This is particularly important because the Black Sea differs from oxygen deficient zones in several important ways. The Black Sea has a sulfidic zone. There are fluxes of reduced species out of the sulfidic zone: reduced S, ammonium, reduced manganese, and methane among other reduced species. Additionally, the zone above the sulfide is suboxic rather than anoxic. Oxygen deficient zones, on the other hand, are mid-water zones that have oxygenated water below them. They are truly anoxic (Revsbech et al., 2009). They don't have fluxes of reduced species entering them. In fact, ammonium is usually below detection (Widner, Fuchsman, et al., 2018; Widner, Mordy, et al., 2018). In the Black Sea, the flux of ammonium from the sulfidic zone determines the importance of anammox and its place in the water column. A comparison of depth profiles of anammox bacteria, ammonium flux and N2 gas can be seen in Fuchsman et al 2012a. Linkage between aerobic ammonium oxidation of the upward flux of ammonium and anammox can be found in (Lam et al., 2007) Additionally, the Black Sea is known to have an organic matter maximum in the redoxycline and quite a bit of information is known about this maximum. (see below) The authors have data from the Black Sea and they need to be more focused on understanding that unique system.

What can this float data tell us about the Black Sea? Are the particle maxima larger on the edges than in the middle of the Sea? Is there a correlation between euphotic zone particles and size of the suboxic zone particle max? Are particle flux events correlated to a season? Not that these particular questions need to be answered. At the moment, this paper tells us things that we already know (there is a particle maximum between 3 uM oxygen and 10 uM sulfide), but I think it could easily tell us more.

I think it is important to note that manganese oxides are quite abundant in the Black Sea and that manganese oxides are also particles in the 0.2-20 um size range. Particle backscattering detects particles of all kinds. In the Black Sea, both ammonium and Mn2+ have an upward flux from the sulfide zone. Anammox bacteria use the ammonium flux to produce N2 gas (Fuchsman et al 2012a) and the Mn2+ is oxidized to manganese oxides under very low oxygen levels in the same zone (Clement et al., 2009). Thus excess N2 gas and manganese oxides are correlated. That correlation is not due to causation however. The authors need to consider how the manganese oxides affect their results. See (Clement et al., 2009; Dellwig et al., 2010; Yakushev et al., 2009) for more information about manganese oxides in the Black Sea.

The ability to detect particles in the water is not a measurement that only exists on floats, but is also present on CTD packages. Thus let us look at a station in the Black Sea where we have all the relevant data—the Western Gyre station in 2005. Here the maximum in organic C associated with microbes is found at sigma theta 16.3 (Figure 1 Fuchsman et al 2011). The maximum in anammox bacteria at the same cruise/station is at sigma theta 16.0-16.1 and the maximum in biologically produced N2 gas is at sigma theta 15.9-16.3 (Fuchsman et al 2012a Figure 1d). The maximum in MnO2 is at sigma theta 15.85 (Fuchsman et al 2011 Figure 6c). There is a small minimum in transmission from 15.8-15.85. The transmission signal corresponds to the manganese oxide peak not the peak in anammox bacteria or organic matter.

However, that particular station didn't have a large organic matter signal in the redoxycline. From looking at the authors' data, I would guess that they often see the organic matter maximum in the redoxycline. The organic matter maximum in the Black Sea redoxycline is from S oxidizing bacteria, which may or may not be autotrophic denitrifiers (Glaubitz et al., 2010; Kirkpatrick et al., 2018). These organic matter maxima can be dominated by S utilizing autotrophic denitrifiers of the genus Sulfurimonas (Kirkpatrick et al., 2018 Figure 7). And thus they could be involved in N2 production, but it has not been proven. Some useful papers about the organic matter maximum in the redoxycline of the Black Sea (Coban-Yildiz et al., 2006; Ediger et al., 2019; Glaubitz et al., 2010; Yilmaz et al., 2006).

Though anammox and denitrification are very important biogeochemically, they aren't actually the most abundant bacteria found in the Black Sea or oxygen deficient zones. In the ETNP oxygen deficient zone, anammox bacteria reached 10% of the community and complete denitrifiers reach ∼5% in the water and 14% of the community on particles (Fuchsman et al., 2017). The most abundant bacteria in oxygen deficient zones, by far, are nitrate reducing SAR11, reaching 60% of the community (Fuchsman et al., 2017; Tsementzi et al., 2016). In the Black Sea, once again SAR11 are the most abundant bacteria (Fuchsman et al., 2011 Figure 2). The SAR11 cannot make N2 gas. They just reduce nitrate to nitrite. I am just trying to note that for heterotrophic denitrifiers and anammox, the authors are using a bulk measurement to look for changes in bacteria that are rarely more than 10% of the community.

Thus, in the Black Sea, I think the assumption that the particle layer represents anammox and heterotrophic denitrifiers is not ideal. First, there are high concentrations of particulate metals in the Black Sea, particularly mananese oxides. Second, the organic matter maximum in the redoxycline is from S oxidizers. Some of these S-oxidizers may be autotrophic denitrifiers. Some aren't. Thus I think the way the particle maximum is talked about in the paper needs to be shifted. Additionally all this information should be in the introduction and discussion.

Specific Comments:

Was the oxygen data from the floats calibrated? See the work of Seth M. Bushinsky to understand the importance of calibration. This information is glossed over in the methods. I think that in previous float work in the Black Sea, scientists used the sulfide zone as a zero to at least track the drift of the oxygen optode over time. Also, it would be good to have a detection limit for all the different float sensors. Bushinsky et al 2016 Limnology and Oceanography Methods

Line 8: This sentence is not accurate as written.

Line 22-23: I am confused what this sentence is trying to say. I note that N2 gas concentrations can be between 400 and 500 microM in the water due to abiotic gas exchange of N2 from the atmosphere. So the authors really mean to say N2 production not concentration. The use of the word respectively in line 23 implies that denitrification is 20% of N2 production and anammox is 40%. Rather, I think the authors are talking about how 20-40% of N2 production occurs in the water column as opposed to in the sediments. The best citation for this is (DeVries et al., 2013).

Line 25: perhaps "where the bacteria that mediate the process mainly reside"

Line 26-27: I am confused as to the meaning of this sentence? Are the authors trying to say that 90% of the N2 production occurred in the upper ODZ? Perhaps it would be better to say that 90% of N2 production occurred in the upper 50 meters of the ODZ. Additionally, one should either say N2 production or N loss. N loss refers to the loss of nutrients. The N2 is produced not lost. I also note that anammox rates are not always highest at the top of the ODZ. See (De Brabandere et al., 2014)

Paragraph 1: I am having issues with oxygen deficient zones being called suboxic. The deficient part of oxygen deficient zone implies that the system is anoxic. No oxygen. The word was coined to differentiate these anoxic systems from suboxic systems which are called oxygen minimum zones.

Line 93: The best citation is (Dalsgaard et al., 2014). The authors do cite this paper later. To be consistent it should be noted here as well.

Line 121-122: This sentence needs clarification for two reasons. The authors are comparing depth and density. The Black Sea is much more consistent in density space than depth. It would be good to give the density range as well as the depth in line 121. Additionally, the authors compare a depth where sulfide is 11 uM to a depth where it is 10 nM. It is not surprising that the 11 uM depth is deeper than the 10 nM depth. That's
an order of magnitude different in concentration. What is the HS- detection limit of the float?

Lines 133-148: The particle layer is between 3 uM oxygen and 11 uM sulfide. Both manganese oxides, and S oxidizers are also found in this range as well as methane oxidizers (Kirkpatrick et al 2018 Figure 6D)– not just anammox and denitrifiers. It is true however, that lots of microbial activity is occurring in this zone. These processes also could all affected by intrusions of oxygen. Lines 142-144: This is interesting.

Line 150-151: This sentence is confusing. I am glad that the authors acknowledge manganese oxides existence. However, manganese oxides are formed by manganese oxidizing bacteria not by denitrifiers. Perhaps autotrophic denitrifiers and manganese oxides, as concepts, should be separated into two sentences.

Line 171: Are the authors that confident in their oxygen concentrations? This would only be true if the sensors are calibrated. Can the optode see the difference between 0.2 uM and 0 uM??

Line 190: Can you actually differentiate correlations with temperature from correlations with density in these deep layers? There is no biological reason that a change < 0.1 in temperature should matter. However, I think many things, such as sulfide, correlate with temperature in this basin.

Line 235: (Cavan et al., 2018) Line 237: (Margolskee et al., 2019)

3.4 New perspectives for studying N2 losses in suboxic ODZs : This section would be more compelling if the floats measured N2 gas. There is such a device—Reed et al 2018 Deep Sea Research Part I 139: 68-78.

References

Cavan, E. L., Giering, S. L. C., Wolff, G. A., Trimmer, M., & Sanders, R. (2018). Alternative Particle Formation Pathways in the Eastern Tropical North Pacific ' s Biological Carbon Pump. Jounral of Geophysical Research: Biogeosciences, 123, 2198–2211.

https://doi.org/10.1029/2018JG004392

Clement, B. G., Luther, G. W., & Tebo, B. M. (2009). Rapid, oxygen-dependent microbial Mn (II) oxidation kinetics at sub-micromolar oxygen concentrations in the Black Sea suboxic zone. Geochimica et Cosmochimica Acta, 73(7), 1878–1889. https://doi.org/10.1016/j.gca.2008.12.023

Coban-Yildiz, Y., Altabet, M. A., Yilmaz, A., & Tugrul, S. (2006). Carbon and nitrogen isotopic ratios of suspended particulate organic matter (SPOM) in the Black Sea water column. Deep Sea Research Part II: Topical Studies in Oceanography, 53(17–19), 1875–1892. https://doi.org/10.1016/j.dsr2.2006.03.021

Dalsgaard, T., Stewart, F. J., Thamdrup, B., Brabandere, L. De, Revsbech, P., & Ulloa, O. (2014). Oxygen at Nanomolar Levels Reversibly Suppresses Process Rates and Gene Expression in Anammox and Denitrification in the Oxygen Minimum Zone off Northern Chile. MBio, 5(6), e01966-14. https://doi.org/10.1128/mBio.01966-14.Editor

De Brabandere, L., Canfield, D. E., Dalsgaard, T., Friederich, G. E., Revsbech, N. P., Ulloa, O., & Thamdrup, B. (2014). Vertical partitioning of nitrogen-loss processes across the oxic-anoxic interface of an oceanic oxygen minimum zone. Environmental Microbiology, 16, 3041–3054. https://doi.org/10.1111/1462-2920.12255

Dellwig, O., Leipe, T., Ma, C., Glockzin, M., Pollehne, F., Schnetger, B., Yaku-shev, E. V, & Bo, M. E. (2010). A new particulate Mn – Fe – P-shuttle at the redoxcline of anoxic basins. Geochimica et Cosmochimica Acta, 74, 7100–7115. https://doi.org/10.1016/j.gca.2010.09.017

DeVries, T., Deutsch, C., Rafter, P. A., & Primeau, F. (2013). Marine denitrification rates determined from a global 3-D inverse model. Biogeosciences, 10(4), 2481–2496. https://doi.org/10.5194/bg-10-2481-2013

Ediger, D., Murray, J. W., & YÄślmaz, A. (2019). Phytoplankton biomass, primary production and chemoautotrophic production of the Western Black

Sea in April 2003. Journal of Marine Systems, 198(January), 103183. https://doi.org/10.1016/j.jmarsys.2019.103183

Fuchsman, C. A., Devol, A. H., Saunders, J. K., McKay, C., & Rocap, G. (2017). Niche Partitioning of the N cycling microbial community of an offshore Oxygen Deficient Zone. Frontiers in Microbiology, 8, 2384.

Fuchsman, C. A., Kirkpatrick, J. B., Brazelton, W. J., Murray, J. W., & Staley, J. T. (2011). Metabolic strategies of free-living and aggregate-associated bacterial communities inferred from biologic and chemical profiles in the Black Sea suboxic zone. FEMS Microbiology Ecology, 78, 586–603. https://doi.org/10.1111/j.1574-6941.2011.01189.x

Glaubitz, S., Labrenz, M., Jost, G., & Jürgens, K. (2010). Diversity of active chemolithoautotrophic prokaryotes in the sulfidic zone of a Black Sea pelagic redoxcline as determined by rRNA-based stable isotope probing. FEMS Microbiology Ecology, 74(1), 32–41. https://doi.org/10.1111/j.1574-6941.2010.00944.x

Kirkpatrick, J. B., Fuchsman, C. A., Yakushev, E. V., Egorov, A. V., Staley, J. T., & Murray, J. W. (2018). Dark N2 fixation: nifH expression in the redoxcline of the Black Sea. Aquatic Microbial Ecology, 82, 43–58. https://doi.org/10.3354/ame01882

Lam, P., Jensen, M. M., Lavik, G., McGinnis, D. F., Muller, B., Schubert, C. J., Amann, R., Thamdrup, B., & Kuypers, M. M. M. (2007). Linking crenarchaeal and bacterial nitrification to anammox in the Black Sea. Proceedings of the National Academy of Sciences, 104(17), 7104–7109. https://doi.org/10.1073/pnas.0611081104

Margolskee, A., Frenzel, H., Emerson, S., & Deutsch, C. (2019). Ventilation Pathways for the North Pacific Oxygen Deficient Zone. Global Biogeochemical Cycles, 33(7), 875–890. https://doi.org/10.1029/2018GB006149

Revsbech, N. P., Larsen, L. H., Gundersen, J., Dalsgaard, T., Ulloa, O., & Thamdrup, B. (2009). Determination of ultra-low oxygen concentrations in oxygen minimum zones by the STOX sensor. Limnology and Oceanography:Methods, 7, 371–381.

Tsementzi, D., Wu, J., Deutsch, S., Nath, S., Rodriguez-r, L. M., Burns, A. S., Ranjan, P., Sarode, N., Malmstrom, R. R., Padilla, C. C., Stone, B. K., Bristow, L. A., Larsen, M., & Glass, J. B. (2016). SAR11 bacteria linked to ocean anoxia and nitrogen loss. Nature, 536, 179–183. https://doi.org/10.1038/nature19068

Widner, B., Fuchsman, C. A., Chang, B. X., Rocap, G., & Mulholland, M. R. (2018). Utilization of urea and cyanate in waters overlying and within the eastern tropical north Pacific oxygen deficient zone. FEMS Microbiology Ecology, 94(September 2017), fiy138. https://doi.org/10.1093/femsec/fiy138

Widner, B., Mordy, C. W., & Mulholland, M. R. (2018). Cyanate distribution and uptake above and within the Eastern Tropical South Pacific oxygen deficient zone. Limnology and Oceanography, 63, S177–S192. https://doi.org/10.1002/lno.10730

Yakushev, E., Pakhomova, S., Sørenson, K., & Skei, J. (2009). Importance of the different manganese species in the formation of water column redox zones : Observations and modeling. Marine Chemistry, 117(1–4), 59–70. https://doi.org/10.1016/j.marchem.2009.09.007

Yilmaz, A., Coban-Yildiz, Y., Tellikarakoc, F., & Bologa, A. (2006). Surface and midwater sources of organic carbon by photoautotrophic and chemoautotrophic production in the Black Sea. Deep Sea Research Part II: Topical Studies in Oceanography, 53(17–19), 1988–2004. https://doi.org/10.1016/j.dsr2.2006.03.015

---

## Referee Comment (RC2) · Anonymous Referee #2 · 3 Aug 2020

[referee-annotated manuscript omitted]

---

## Author Comment (AC1) · 15 Sep 2020

Dear Dr. Clara A. Fuchsman

Thank you very much for providing your valuable time to review our manuscript. We also thank you for your constructive feedback because it allowed us to improve the original version of the manuscript. Please find below answers and related actions for all your comments and recommendations.

King regards,

Rafael Rasse Hervé Claustre Antoine Poteau

[Figure]

General Comment

In "The suspended small-particles layer in the suboxic Black Sea: a proxy for delineating the effective N2-yielding section" by Rasse et al, the authors analyze particle back scattering, oxygen, HS- and nitrate float data from the Black Sea. The authors can thus delineate the suboxic zone of the Black Sea with the float, and see productivity and export events. The authors assume that the particle back scattering data in the suboxic zone indicates the presence of anammox and heterotrophic denitrifying bacteria. This assumption may be problematic in the Black Sea where it is known that there are high manganese oxide concentrations in the suboxic zone, and it is known that there is an organic matter maximum at the top of the sulfidic zone composed of S oxidizers. However, the data in this paper is useful. The writing just needs to be shifted.

Answer. We agree and understand the main reviewer's concern about the interpretation of the bbp-layer.

Actions taken. We changed the writing at the required sections, and added the information needed to explain the role that other particles (inorganic & biogenic) have on the formation of the bbp-layer. We are confident you will find this revised version satisfactory (see details below).

Main Comment #1

Large issues.

The introduction needs a section at the beginning describing the Black Sea. This is particularly important because the Black Sea differs from oxygen deficient zones in several important ways. The Black Sea has a sulfidic zone. There are fluxes of reduced species out of the sulfidic zone: reduced S, ammonium, reduced manganese, and methane among other reduced species. Additionally, the zone above the sulfide is suboxic rather than anoxic. Oxygen deficient zones, on the other hand, are mid-water

zones that have oxygenated water below them. They are truly anoxic (Revsbech et al., 2009). They don't have fluxes of reduced species entering them. In fact, ammonium is usually below detection (Widner, Fuchsman, et al., 2018; Widner, Mordy, et al., 2018). In the Black Sea, the flux of ammonium from the sulfidic zone determines the importance of anammox and its place in the water column. A comparison of depth profiles of anammox bacteria, ammonium flux and N2 gas can be seen in Fuchsman et al 2012a. Linkage between aerobic ammonium oxidation of the upward flux of ammonium and anammox can be found in (Lam et al., 2007) Additionally, the Black Sea is known to have an organic matter maximum in the redoxycline and quite a bit of information is known about this maximum. (see below) The authors have data from the Black Sea and they need to be more focused on understanding that unique system.

Answer. We agree, the Black Sea is an ecosystem that clearly differs from oxygen deficient zones as we indicated very briefly between lines 215-216 (original manuscript).

Action taken. We removed the phrase oxygen deficient zones (ODZs) from the introduction and sections required. Instead, we used the term: "poorly-oxygenated water masses" (e.g. O2 < 3 uM), which refers to those water masses at which N2 can be produced independently of the biogeochemical mechanisms driving it. Thus, including basins (e.g. Black Sea and Cariaco basin) and oxygen deficient zones (e.g. ETNP, ETSP, and AS).

We also included a "background section" to describe the key biogeochemical processes and associated inorganic-biogenic particles contributing to the formation of the bbp-layer (see more details below). The changes mentioned above are highlighted in yellow in the following lines of the new manuscript:

- The term "poorly-oxygenated water masses": 1, 8, 23, 25, 30, 35, 40, 42, 49, 57, 60, 63, 64, 69. . .161, etc.

- The background section: 71-94.

Main Comment #2

What can this float data tell us about the Black Sea? Are the particle maxima larger on the edges than in the middle of the Sea? Is there a correlation between euphotic zone particles and size of the suboxic zone particle max? Are particle flux events correlated to a season? Not that these particular questions need to be answered. At the moment, this paper tells us things that we already know (there is a particle maximum between 3uM oxygen and 10 uM sulfide), but I think it could easily tell us more.

Answer. We agree, there are many interesting aspects that can be easily explored with our data set. However, such aspects are out of the scope of our study. In addition, we consider that our findings tell more than a single maximum of particles between 3 uM $O_2$ and 10 uM sulphide. This is mainly because they highlight that the bbp-layer can be exploited as a combined proxy to efficiently delineate and track the effective N2 yielding section. The latter was in part demonstrated by the novel and robust bbp vs $NO_3-$ correlations computed with high spatial and temporal resolutions. The data of optical particles maxima are only used as complementary information to support our findings.

Finally, from our point of view, we consider that the key questions would be: how this optically derived layer of suspended small-particles can be exploited – by first time- to improve current estimates of N2 yielding via the growing BGC-Argo float network? For instance, what does this bbp-layer can tell us about how physical forcing drive spatial-temporal changes in the location and thickness of the effective N2 yielding sections of ODZs? What can occur to the bbp-layer and related N2 yielding rates, if $O_2$ is injected from the bottom instead of $Mn^{2+}$, $NH_4+$ or $H_2S$? How can we exploit the latter two aspects to improve oceanic estimates of N2 production? Overall, this work is only a step forward to build the foundations that we need to achieve the main goal of our ongoing research. We hope the reviewer can also understand our perspective.

Action taken. Not actions were taken.
Main Comment #3

I think it is important to note that manganese oxides are quite abundant in the Black Sea and that manganese oxides are also particles in the 0.2-20 um size range. Particle backscattering detects particles of all kinds. In the Black Sea, both ammonium and Mn2+ have an upward flux from the sulfide zone. Anammox bacteria use the ammonium flux to produce N2 gas (Fuchsman et al 2012a) and the Mn2+ is oxidized to manganese oxides under very low oxygen levels in the same zone (Clement et al., 2009). Thus excess N2 gas and manganese oxides are correlated. That correlation is not due to causation however. The authors need to consider how the manganese oxides affect their results. See (Clement et al., 2009; Dellwig et al., 2010; Yakushev et al., 2009) for more information about manganese oxides in the Black Sea.

Answer. We agree with the reviewer. The role of manganese oxides and its link with the suspended small particles layer should be described better.

Action taken. The role of manganese oxides (mainly as MnO2) was included in both the new "background section" and discussion. The changes indicated above are highlighted in yellow in the following lines of the new version: - 71-94, 171-197, and 207-230.

Main Comment #4

[a] The ability to detect particles in the water is not a measurement that only exists on floats, but is also present on CTD packages. Thus, let us look at a station in the Black Sea where we have all the relevant dataâĚŸAĚĞ T the Western Gyre station in 2005. [b & c] Here the maximum in organic C associated with microbes is found at sigma theta 16.3 (Figure 1 Fuchsman et al 2011). The maximum in anammox bacteria at the same cruise/station is at sigma theta 16.0-16.1 and the maximum in biologically produced N2 gas is at sigma theta 15.9-16.3 (Fuchsman et al 2012a Figure 1d). The maximum in MnO2 is at sigma theta 15.85 (Fuchsman et al 2011 Figure 6c). There is a small minimum in transmission from 15.8-15.85. The transmission signal corresponds

to the manganese oxide peak not the peak in anammox bacteria or organic matter. However, that particular station didn't have a large organic matter signal in the redoxycline. From looking at the authors' data, I would guess that they often see the organic matter maximum in the redoxycline. The organic matter maximum in the Black Sea redoxycline is from S oxidizing bacteria, which may or may not be autotrophic denitrifiers (Glaubitz et al., 2010; Kirkpatrick et al., 2018). These organic matter maxima can be dominated by S utilizing autotrophic denitrifiers of the genus Sulfurimonas (Kirkpatrick et al., 2018 Figure 7). And thus they could be involved in N2 production, but it has not been proven. Some useful papers about the organic matter maximum in the redoxycline of the Black Sea (Coban-Yildiz et al., 2006; Ediger et al., 2019; Glaubitz et al., 2010; Yilmaz et al., 2006).

[d] Though anammox and denitrification are very important biogeochemically, they aren't actually the most abundant bacteria found in the Black Sea or oxygen deficient zones. In the ETNP oxygen deficient zone, anammox bacteria reached 10% of the community and complete denitrifiers reach _5% in the water and 14% of the community on particles (Fuchsman et al., 2017). The most abundant bacteria in oxygen deficient zones, by far, are nitrate reducing SAR11, reaching 60% of the community (Fuchsman et al., 2017; Tsementzi et al., 2016). In the Black Sea, once again SAR11 are the most abundant bacteria (Fuchsman et al., 2011 Figure 2). The SAR11 cannot make N2 gas. They just reduce nitrate to nitrite. I am just trying to note that for heterotrophic denitrifiers and anammox, the authors are using a bulk measurement to look for changes in bacteria that are rarely more than 10% of the community.

Answers.

[a] CTD. We agree, CTD packages provide very valuable information about the physical, optical, and biogeochemical properties of the ocean. For instance, they generate variables that cannot be measured by the current float sensors as well as of those variables used to calibrate them. However, they can only provide a snapshot for a given time of the sampling day, while discrete samples are often collected with poor vertical

resolution. We thus consider that alternative methods need to be developed to complement CTD packages, and ultimately better understand biogeochemical cycles in poorly oxygenated regions as proposed here (please see also our answer in the main comment #1, section 4.4. e.g. and refs: Chai et al., 2020; Claustre et al., 2020; Martin et al., 2020).

Action taken. No actions were taken.

[b] Comparison between the vertical profiles of the particles measured by CTD and floats. We reviewed all figures and articles cited by the reviewer. Again, this is a very interesting description of the vertical profiles of N2, and particles using data collected via CTD packages. Even though we used the profile of March 2005 to highlight the qualitative relationship between N2 excess and optical particles only in the effective N2 production section; we consider these data are not the most suitable to compare the vertical profiles of particles derived from CTD and floats (including those from Coban-Yildiz et al., 2006; Ediger et al., 2019; Glaubitz et al., 2010; Yilmaz et al., 2006). The main reason is that the vertical profiles of particles cited by the reviewer are representative of the region impacted by the Bosporus plume. In this region, lateral advection via the Bosporus plume drives a maximum of particles $\sim$ 16.3 kg m-3 (or higher) because it fuels chemoautotrophic activities by injecting NO3- (e.g. $\sim$ 700 m depth, Stanev et al. 2017). Thus, these CTD-profiles of particles must be similar to those excluded in our analysis (e.g. see our Figure 2 between May-June). Note that we focus on the in-situ 1D processes driving the local formation of the bbp-layer (this was indicated between lines 83-84 of the original manuscript). Thus, our data set is more representative of the profile described in Figures 1c-1d of Lam et al. 2017 indicated below. Also note that our bbp maxima are located between the isopycnals 15.79 kg m-3 and 16.3 kg m-3 (Figure 2, and Figure 3g of the manuscript).

Action taken. No actions were taken.

[c, d] Vertical profile of biogenic and organic particles. We thank the reviewer for this

didactic and constructive comment.

Again, we agree. There are different biogenic and inorganic particles that contribute to the bbp-layer, as we briefly recognize between lines 196-197 of the original manuscript . We only indicate that annamox-denitrifying bacteria are at least partial contributors of the bbp-layer because they should have the major contribution to N2 yielding and removal rates of NO3-. Hence, we did not suggest that such bacteria are the main microbial component (or organic particles) that contribute to the bbp-layer. It is clear that bbp signal is a black box that cannot be attributed to a single type of organic-inorganic particles.

As mentioned above, we are only trying to highlight two aspects: (1) the bbp-layer is systematically delineating and tracking the effective N2 yielding section, independently of the physical (e.g. NH4+, Mn2+, H2S pumped from the sulfidic zone) and biogeo-chemical mechanisms driving both N2 yielding and the small-particles content, and (2) similar results should also be expected for the bbp-layer of the ODZ because this layer must be mainly due to the microbial communities involved in N2 yielding (see also the answer of the main comment # 1, section 4.4, and refs: Martin and Knauer, 1984; Johnson et al., 1996; Lewis and Luther, 2000).

Action taken. We added information about the other inorganic and biogenic particles that contribute to the formation of the bbp-layer, and modified the writing at the required sections. Such changes are highlighted in yellow in the following lines of the new manuscript: - 71-94, 171-197, and 207-230.

Main Comment #5

Thus, in the Black Sea, I think the assumption that the particle layer represents anam-mox and heterotrophic denitrifiers is not ideal. First, there are high concentrations of particulate metals in the Black Sea, particularly mananese oxides. Second, the organic matter maximum in the redoxycline is from S oxidizers. Some of these S-oxidizers may be autotrophic denitrifiers. Some aren't. Thus I think the way the particle maximum is

talked about in the paper needs to be shifted. Additionally all this information should be in the introduction and discussion.

Answer. We agree

Action taken. We changed how the maximum of optical particles is described (see all details above and in the new version).

Specific comments [ES].

ES comment # 1

Was the oxygen data from the floats calibrated? See the work of Seth M. Bushinsky to understand the importance of calibration. This information is glossed over in the methods. I think that in previous float work in the Black Sea, scientists used the sulphide zone as a zero to at least track the drift of the oxygen optode over time. Also, it would be good to have a detection limit for all the different float sensors. Bushinsky et al 2016 Limnology and Oceanography Methods

Answer. Yes, the optodes 43330 are multi-point calibrated. The oxygen data is adjusted and corrected following the Argo quality control manual for dissolved oxygen concentration as cited in the manuscript (Thierry et al. 2018).

Action taken. We added both the range of O2 concentrations that can be measured by the Optode sensors and their accuracy. This information is reported by the manufacturer. In addition, we cited the work related to the O2 Optode Drift correction applied to the sensors (Bittig, and Körtzinger. 2015).

ES comment # 2

Line 8: This sentence is not accurate as written.

Answer. Agree

Action taken. We modified this sentence.

[Figure]

ES comment # 3

Line 22-23: I am confused what this sentence is trying to say. I note that N2 gas concentrations can be between 400 and 500 microM in the water due to abiotic gas exchange of N2 from the atmosphere. So the authors really mean to say N2 production not concentration. The use of the word respectively in line 23 implies that denitrification is 20% of N2 production and anammox is 40%. Rather, I think the authors are talking about how 20-40% of N2 production occurs in the water column as opposed to in the sediments. The best citation for this is (DeVries et al., 2013).

Answer. We did not understand this observation because N2 concentration is not mentioned in this sentence. We only indicate that N2 is mainly produced via anammox-denitrification.

Action taken. We modified this sentence and removed the word "respectively".

ES comment # 4

Line 25: perhaps "where the bacteria that mediate the process mainly reside"

Answer. OK.

Action taken. Not action was taken because this is only a semantic issue.

ES comment # 5

Line 26-27: I am confused as to the meaning of this sentence? Are the authors trying to say that 90% of the N2 production occurred in the upper ODZ? Perhaps it would be better to say that 90% of N2 production occurred in the upper 50 meters of the ODZ. Additionally, one should either say N2 production or N loss. N loss refers to the loss of nutrients. The N2 is produced not lost. I also note that anammox rates are not always highest at the top of the ODZ. See (De Brabandere et al., 2014)

Answer. OK.

Action taken. The term ODZs was removed from the introduction. We now use the term "shallower poorly-oxygenated water masses" and replaced the word "loss" by "yielding" throughout the manuscript.

ES comment # 6

Paragraph 1: I am having issues with oxygen deficient zones being called suboxic. The deficient part of oxygen deficient zone implies that the system is anoxic. No oxygen. The word was coined to differentiate these anoxic systems from suboxic systems which are called oxygen minimum zones.

Answer. We agree, the application of the term suboxic is not accurate and very confusing because the $O_2$ levels used to define it can potentially overlap with hypoxic ($O_2$ < 60 uM, e.g. Stramma et al. 2008 ) and anoxic conditions ($O_2$ < 1-2 uM). Example, for the Black Sea, the suboxic zone is set between $O_2$ levels ranging between 1.8 uM and 39 uM (Fuschman et al., 2011; Stanev et al., 2018 and references therein).

Action taken. We now use the term "poorly oxygenated water masses" to describe the section where $N_2$ is effectively produced. We also introduced the term, "chemical zones" because this is more suitable (or accurate) to explain the biogeochemical reactions that drive $N_2$ yielding and the small-particle content in the poorly oxygenated water masses (Canfield and Thamdrup, 2009).

The changes indicated above are highlighted in yellow in the following lines of the new manuscript: - Lines: 71-94, 171-197, and 207-230.

ES comment # 7

Line 93: The best citation is (Dalsgaard et al., 2014). The authors do cite this paper later. To be consistent it should be noted here as well.

Answer. OK

Action taken. We cited Dalsgaard et al., 2014 as well.

ES comment # 8

Line 121-122: This sentence needs clarification for two reasons. The authors are comparing depth and density. The Black Sea is much more consistent in density space than depth. It would be good to give the density range as well as the depth in line 121. Additionally, the authors compare a depth where sulfide is 11 uM to a depth where it is 10 nM. It is not surprising that the 11 uM depth is deeper than the 10 nM depth. That's an order of magnitude different in concentration. What is the HS- detection limit of the float?

Answer. OK

Action taken. We reported the detection limit of HS-, as well as the ranges of depth and density. We also modified the sentence between lines 121-122 of the old manuscript.

ES comment # 9

Lines 133-148: The particle layer is between 3 uM oxygen and 11 uM sulfide. Both manganese oxides, and S oxidizers are also found in this range as well as methane oxidizers (Kirkpatrick et al 2018 Figure 6D)– not just anammox and denitrifiers. It is true however, that lots of microbial activity is occurring in this zone. These processes also could all affected by intrusions of oxygen. Lines 142-144: This is interesting.

Answer. OK

Action taken. This sentence was modified. We included the presence of other bacteria and MnO2, and described their links with O2 and NO3- levels according to the case. Such changes are highlighted in yellow in the following lines of the new manuscript: -Lines: 71-94, and 207-230.

ES comment # 10

Line 150-151: This sentence is confusing. I am glad that the authors acknowledge manganese oxides existence. However, manganese oxides are formed by manganese

oxidizing bacteria not by denitrifiers. Perhaps autotrophic denitrifiers and manganese oxides, as concepts, should be separated into two sentences

Answer. OK

Action taken. This sentence was removed.

ES comment # 11

Line 171: Are the authors that confident in their oxygen concentrations? This would only be true if the sensors are calibrated. Can the optode see the difference between 0.2 uM and 0 uM??

Answer. Yes. According to the manufacturer (https://www.aanderaa.com/media/pdfs/d378_aanderaa_oxygen_sensor_4330_4330f.pdf), the Optode sensors can measure O2 concentrations between 0 to 1000 uM with an accuracy of 1.5%. Thus, in theory, we should be able to see the difference between 0.2 uM and 0 uM. However, the minimum value of O2 measured by the sensor and used in our calculation was 0.22 uM. Thus, we cannot confirm (or regret) that such sensors can see such difference.

Action taken. We added the range of O2 concentrations and accuracy of the Optode sensor reported for the manufacturer. Finally, we indicated that 0.22 uM was the minimum value of O2 measured by the sensor, and suggested that the O2 level can be also lower (see line 219 in the new one)

ES comment # 12

Line 190: Can you actually differentiate correlations with temperature from correlations with density in these deep layers? There is no biological reason that a change < 0.1 in temperature should matter. However, I think many things, such as sulfide, correlate with temperature in this basin.

Answer. Correlations between bbp vs T were computed between two reference isopycnals delineating the two sub-zones of the bbp-layer. The same principle was applied for bbp vs NO3 –. Thus, these correlations are consistently found in the water layers of the two sub-zones (or "chemical zones") defined for the bbp-layer. which ultimately validates -in part- our main hypothesis. However, our data cannot explain why such correlation is found.

Action taken. The following sentence was added:

Finally, more information is needed to investigate the physical and/or biogeochemical processes driving the correlation between the increase rates of T, and declines rates of NO3- in the first sub-zone. The former is, however, out of the scope of our study. (lines 238-240 in the new version)

ES comment # 13

Line 235: (Cavan et al., 2018) Line 237: (Margolskee et al., 2019)

Answer. OK

Action taken. We cited Margolskee et al., 2019 and Cavan et al., 2017 because the latter is more suitable for this context (see also Rasse and Dall'Olmo, 2019 for the ETNA-OMZ case; http://dx.doi.org/10.1029/2019GB006305).

ES comment # 14

3.4 New perspectives for studying N2 losses in suboxic ODZs : This section would be more compelling if the floats measured N2 gas. There is such a deviceâĚŸA ĚĞTReed et al 2018 Deep Sea Research Part I 139: 68-78.

Answer. OK

Action taken. We cited Reed et al., 2018 and indicated that N2 can be measured by BGC-floats.

References not cited in the new version of the manuscript

Chai, F., Johnson, K. S., Claustre, H., Xing, X., Wang, Y., Boss, E., ... & Sutton, A. (2020). Monitoring ocean biogeochemistry with autonomous platforms. Nature Reviews Earth & Environment, 1-12. https://doi.org/10.1038/s43017-020-0053-y

Claustre, H., Johnson, K. S., & Takeshita, Y. (2020). Observing the global ocean with biogeochemical-Argo. Annual Review of Marine Science, 12, 23-48. https://doi.org/10.1146/annurev-marine-010419-010956

Stramma, L., Johnson, G. C., Sprintall, J., & Mohrholz, V. (2008). Expanding oxygen-minimum zones in the tropical oceans. science, 320(5876), 655-658. DOI: 10.1126/science.1153847

Martin, A., Boyd, P., Buesseler, K., Cetinic, I., Claustre, H., Giering, S., ... & Robinson, C. (2020). The oceans' twilight zone must be studied now, before it is too late. doi: 10.1038/d41586-020-00915-7

[Figure]

[Figure]

[Figure]

Fig. 1. Vertical distribution of inorganic nitrogen (a), O$_2$ and sulfide (b), light transmission, particulate MnO$_x$ and total reduced Mn (c), and anammox bacterial abundance and $^{15}$N$_2$ production rates (d).

**Fig. 1.**

---

## Author Comment (AC2) · 15 Sep 2020

Dear reviewer,

Thank you very much for spending part of your valuable time reviewing our manuscript. We also thank you for your constructive feedback because it allowed us to improve the original version of the manuscript. Below, you will find our answers and actions taken for each of your comments.

King regards,

Rafael Rasse Hervé Claustre Antoine Poteau

Comment #1

What is the typical depths ??

Are these depths vary among different ODZs?

Answer. OK

Action taken. This sentence was modified. We indicated the depths at which this layer can be found. This information is based on data from The Black Sea (this study), and the ODZs of the Arabian Sea and ETSP (Whitmire et al. 2009; Wojtasiewicz et al. 2018).

Comment #2

Are these factors listed in order of their importance?

Answer. According to the literature, we consider this is the most likely order.

Action taken. No actions were taken.

Comment #3

Will not the chemical composition, salinity and temperature of water column would also matter for resultant optical visibility / abundance of anammox and denitrifying bacteria ??

Answer. Organic matter composition should be key driving the microbial activity (e.g. anammox and denitrifying bacteria, e.g. Van Mooy et al. 2002) but this not be critical for our case (see line 165 in the old manuscript and the cited work). We mentioned an array of chemical variables (levels O2, NO3, and HS, OM) at the line 34 of the old version. We don't have information about salinity but T can affect their activity in sediments (e.g. Rysgaard et al. 2004; Canion et al., 2014).

Action taken. No actions were taken.

Comment #4

Here authors are attempting to investigate measured Bbp layer (absorption ?) with chemical parameters such as O2, NO3 H2S and N2 produced.....all chemical parameters is there any way to provide Bbp thickness and its absorption correlation with actual density of microbial mass...(just wondering samples collected on filters??)

Answer. We did not have such data .

Action taken. No actions were taken.

Comment #5

How much thick it is?

Answer. It can be highly variable with time, and between ODZs and anoxic basins. Please see section 4.1, where we indicate the thickness of the bbp-layer for the case of the Black Sea.

Action taken. No actions were taken.

Comment #6

Suppose this factor is negligible in some locations ??

Answer. Please, see how the ventilation of subsurface O2 defines the characteristics of the bbp-layer and how we used such information to explain what are the main particles contributing to its formation (e.g. section 4.2).

Action taken. No actions were taken.

Comment #7

why ? what is another factor for second sub-zone?

Answer. This is related to the biogeochemical processes that control the content of suspended small particles and N2 excess in the chemical zones of the poorly-oxygenated water masses. This is better described in the new version of the manuscript.

Action taken. We included a new "background section" to describe the key biogeo-chemical processes and associated inorganic-biogenic particles contributing to the for-mation of the bbp-layer. The interlinks among biogeochemical processes, and the vertical profiles of small-particles and N2 excess are described in the discussion as well.

These changes are highlighted in yellow in the following lines of the new version: - 71-94, 171-197, and 207-230.

Comment #8: Sentences highlighted in yellow without suggestions

- of chl and bbp and due to particle

Answer. Both spikes are due to particles-aggregates. We thus consider this sentence is OK

Action taken. No actions were taken.

- o free-living bacteria (0.2-2 $\mu$m), and those associated with small-suspended particles (> 2-20 $\mu$m).

Answer. These ranges of particles size are explained in the introduction.

Action taken. No actions were taken.
* * *
- hypothesized

- Optical proxies of tiny particles can be applied as an alternative approach to assess the vertical distribution of N2-yieldingmicrobial communities in upper suboxic ODZs

- particle content inferred from bbp and N2 produced by microbial communities are at least qualitatively correlated microbial communities in upper suboxic ODZs

- bbp and O2 can be exploited as a combined proxy for defining the N2-producing section of the suboxic Black Sea

- fluorescence and total backscattering were converted into Chlorophyll concentration (chl) and particle backscattering (bbp) following standard protocols

- HS- was not used to delimit the bottom of this zone because the maximum concentration of H2S that denitrifying and anammox bacteria tolerate is not well established.

- NO3- and O2 are two of the key factors that modulate the presence of denitrifying and anammox bacteria

- bbp-layer is partially composed of N2-yielding microbial communities such as anammox and denitrifying bacteria.

- bbp-layer is at least partially composed of anaerobic microbial communities involved in the production of N2

Answer. OK

Action taken. The sentences above were modified.
* * *
—-

Comment #9: Other sentences highlighted in yellow without suggestions

- How key drivers of anammox-denitrifying bacteria dynamics impact on the vertical distribution of bbp and the thickness of the bbp-layer.

- Optical proxies of tiny particles can be applied as an alternative approach to assess the vertical distribution of N2-yielding.

- Slightly sulfidic conditions of the deepest isopycnal at which anammox bacteria can be still recorded.

- It is still debated whether the oceanic nitrogen cycle is in balance or not.

Answers. Because it is not specified what are the issues with the sentences above; we assumed that these are only semantic issues.

Action taken. No actions were taken.

References.

Canion, A., Kostka, J. E., Gihring, T. M., Huettel, M., Van Beusekom, J. E. E., Gao, H., ... & Kuypers, M. M. (2014). Temperature response of denitrification and anammox reveals the adaptation of microbial communities to in situ temperatures in permeable marine sediments that span 50? in latitude. Biogeosciences, 11(2), 309.

Rysgaard, S., Glud, R. N., Risgaard-Petersen, N., & Dalsgaard, T. (2004). Denitrification and anammox activity in Arctic marine sediments. Limnology and Oceanography, 49(5), 1493-1502.

Whitmire, A. L., Letelier, R. M., Villagrán, V., and Ulloa, O.: Autonomous observations of in vivo fluorescence and particle backscattering in an oceanic oxygen minimum zone, Opt. Express, 17(24), 21, 992–22,004. https://doi.org/10.1364/OE.17.021992, 2009.

Wojtasiewicz, B., Trull, T. W., Bhaskar, T. U., Gauns, M., Prakash, S., Ravichandran, M., and Hardman‐Mountford, N. J.: Autonomous profiling float observations reveal the dynamics of deep biomass distributions in the denitrifying oxygen minimum zone of the Arabian Sea, J. Mar. Syst., https://doi.org/10.1016/j.jmarsys.2018.07.002, 2020.

Van Mooy, B. A., Keil, R. G., & Devol, A. H. (2002). Impact of suboxia on sinking particulate organic carbon: Enhanced carbon flux and preferential degradation of amino acids via denitrification. Geochimica et Cosmochimica Acta, 66(3), 457-465.

---

## Author Response (AR2)

[revised manuscript text omitted]

2006.

**General Comment**

I thank the authors for expanding the introduction and discussion to include manganese oxides as a source of particles. I think this addition is important to the paper. However, the authors still need to add in references to previous work on the organic matter maximum at the redox interface In the Black Sea. Just one or two sentences acknowledging this previous work seems important since this organic matter maximum is the focus of the present paper. Important papers about the organic matter maximum in the Black Sea redoxycline: (Coban-Yildiz et al., 2006; Yilmaz et al., 2006; Glaubitz et al., 2010; Ediger et al., 2019) Perhaps add these new sentences around line 29 of the introduction.

First, we would like to thank Dr. Clara A. Fuschman for her constructive, positive, and accurate feedback. The latter allowed us to improve the original version of this manuscript. We are confident that you will find this revised version satisfactory.

**Answer.** We agree, Epsilonproteobacteria have the most significant contribution to the formation of organic particles *mainly* in the sulfidic zone.

**Action taken**. We added the information related to the role of such bacteria in the formation of organic particles and $N_2$ yielding. This information is in the section 2.0 between lines 83-88 of the last version (text highlighted in green). Related references were also included.

**Comment #1**
Line 31: N2 yielding bacteria—the noun is necessary
Line 77: N2 yielding bacteria

**Answer.** OK
**Action taken**. The noun $N_2$ yielding bacteria was added (text highlighted in green at lines 31 and 78, respectively)

**Comment #2**
Poorly oxygenated isn't a scientific term. I think the term you are looking for is suboxic.

**Answer**. Oxygen-poor waters is a scientific term that was already used by Stramma et al. 2008 (abstract), 2010 (Introduction, 2nd paragraph). We thus consider that the latter term is equivalent to the one used here (poorly-oxygenated).

**Action taken**. To be consistent with what is reported in the literature, we changed the terms poorly-oxygenated by oxygen-poor throughout the manuscript. These changes are highlighted in green at the respective lines of the revised manuscript.

**Comment #3**
**Line 174:** The epsilonproteobacteria Sulfurimonas is one of the most important sulfur oxidizers in the Black Sea. See (Glaubitz et al., 2010) also (Kirkpatrick et al., 2018) figure 7. This epsilon proteobacteria is likely an autotrophic denitrifier (Fuchsman et al., 2012). Might be a better sulfur oxidizer to single out than SUP05—or name them both.

**Answer.** OK
**Action taken**. We included the information requested between lines 178-179 of the revised manuscript (highlighted in green).

**Comment #4**
Line 28: the fact that some SAR11 can reduce nitrate is from (Tsementzi et al., 2016). I know you cite
this paper later, but it would be good to cite it here too.
**Answer. OK**
**Action taken**. The reference was included at the line specified.
**Comment #5**
Line 312: Actually, your results do not suggest that the particle layer is due to the list of bacteria. Previous
work suggests this. For example (Glaubitz et al., 2010) or (Kirkpatrick et al., 2018). You assume that this
is the case. Please rephrase.
**Answer.** OK
**Action taken**. This sentence was rephrased.